# Pd(II)/Pd(IV) redox shuttle to suppress vacancy defects at grain boundaries for efficient kesterite solar cells

Jinlin Wang[1,2,7], Jiangjian Shi [1,7], Kang Yin[1,2,7], Fanqi Meng [3,7], Shanshan Wang[4], Licheng Lou[1,2], Jiazheng Zhou[1,2], Xiao Xu[1,2], Huijue Wu[1], Yanhong Luo [1,2,5], Dongmei Li [1,2,5] ✉, Shiyou Chen [4] ✉ & Qingbo Meng [1,5,6] ✉

Charge loss at grain boundaries of kesterite $Cu_2ZnSn(S, Se)_4$ polycrystalline absorbers is an important cause limiting the performance of this emerging thin-film solar cell. Herein, we report a Pd element assisted reaction strategy to suppress atomic vacancy defects in GB regions. The Pd, on one hand in the form of $PdSe_x$ compounds, can heterogeneously cover the GBs of the absorber film, suppressing Sn and Se volatilization loss and the formation of their vacancy defects (i.e. $V_{Sn}$ and $V_{Se}$), and on the other hand, in the form of Pd(II)/Pd(IV) redox shuttle, can assist the capture and exchange of Se atoms, thus contributing to eliminating the already-existing $V_{Se}$ defects within GBs. These collective effects have effectively reduced charge recombination loss and enhanced p-type characteristics of the kesterite absorber. As a result, high-performance kesterite solar cells with a total-area efficiency of 14.5% (certified at 14.3%) have been achieved.

Kesterite solar cell has emerged as one of the most promising thin-film photovoltaics since the absorber, $Cu_2ZnSn(S, Se)_4$ (CZTSSe), is a nontoxic material composed of earth-abundant elements and has excellent thermodynamic stability and moreover this cell technology is compatible with current thin-film photovoltaics industry[1–4]. This solar cell has garnered increasing research attention in recent years and efficiency breakthroughs have been successively reported, with the most recent achievements successively reaching 13% and 14% using environmentally-friendly solution methods[5–7]. Nonetheless, these cell efficiencies still fall significantly short of its Shockley-Queisser limit. This efficiency gap, particularly exhibiting as significant deficits in the open-circuit voltage ($V_{OC}$), are primarily caused by intricate defects in CZTSSe absorbers and the induced charge non-radiative recombination loss.

For polycrystalline CZTSSe, the complex phase evolution processes, the coexistence of secondary phases, and the disorder of multinary elements are considered as main causes of defects and charge loss[8–11]. In previous studies, a variety of efforts have been paid to relieve these issues and particularly to suppress intrinsic point defects within the bulk grain interiors (GI)[12–16], which has made considerable contribution to the efficiency improvement of the cell. In addition to these widely concerned issues, Hao et al[17]. recently highlighted that the grain boundary (GB) within CZTSSe absorbers actually played a more substantial role in influencing charge recombination velocity and charge loss. Several factors contribute to this phenomenon regarding GBs. Firstly, GBs typically exhibit a higher degree of structural distortion and atomic disorder, leading to the formation of various types of defects, such as Se-Se dimers[18–20]. Secondly,

[1]Beijing National Laboratory for Condensed Matter Physics, Institute of Physics, Chinese Academy of Sciences (CAS), Beijing 100190, P. R. China. [2]School of Physical Sciences, University of Chinese Academy of Sciences, Beijing 100049, P. R. China. [3]School of Materials Science and Engineering, Peking University, Beijing 100871, P. R. China. [4]School of Microelectronics, Fudan University, Shanghai 200433, P. R. China. [5]Songshan Lake Materials Laboratory, Dongguan 523808, P. R. China. [6]Center of Materials Science and Optoelectronics Engineering, University of Chinese Academy of Sciences, Beijing 100049, P. R. China. [7]These authors contributed equally: Jinlin Wang, Jiangjian Shi, Kang Yin, Fanqi Meng. ✉e-mail: dmli@iphy.ac.cn; chensy@fudan.edu.cn; qbmeng@iphy.ac.cn

conductive secondary phases such as $Cu_xSe$ and $SnSe_x$ tend to segregate at GBs[21–23], creating current shunting pathways within these boundaries. Thirdly, during the later stages of the selenization process, the low-Se-pressure reaction environment struggles to maintain CZTSSe crystals in a stable state, leading to gradual surface decomposition[24,25].

Due to the aforementioned reasons, precise regulation of defects at GBs of CZTSSe absorbers has become an essential prerequisite for further enhancing the efficiency of kesterite solar cells. Researchers have attempted several strategies for overcoming issues such as the detrimental Se-Se dimers and the segregation of secondary phases, by using alkali metal incorporation[26,27], surface etching[22,28], composition control[29–31], and optimization of reaction pathways[6,7]. These efforts have yielded promising results sequentially; however, the previously generally believed positive effect of GB on charge transport[32–34] and the intricate growth process of CZTSSe polycrystalline films at elevated temperatures has led to a dearth of investigations into the characteristics of defects situated in the GB regions and their impacts. Consequently, no matter from physics understanding or from the material engineering, the regulation of defects within the GBs of CZTSSe remains a substantial challenge in this field.

Herein, we propose that vacancy defects caused by the inevitable and irreversible volatilization loss of Se or Sn elements at GBs of the CZTSSe film at elevated temperatures are important origins for the charge loss. To overcome this issue directly related to the thermodynamic properties of the constituent elements of CZTSSe, we started from the perspective of controlling the volatilization kinetics and developed a local reaction strategy leveraging palladium (Pd). In this strategy, non-volatile $PdSe_x$ compounds were introduced to heterogeneously fill into and cover GBs of the CZTSSe film by taking advantage of the spontaneous segregation of Pd from the CZTSSe GIs. This local covering can effectively suppress the element volatilization loss and the formation of vacancy defects. Moreover, Pd(II)/Pd(IV) is a redox shuttle, which can capture vaporized Se from the reaction environment and subsequently supply to the CZTSSe absorber, further diminishing vacancy defects. These effects enable the fabrication of low-defect-density CZTSSe absorbers and consequently achieve the highest certified efficiency of 14.3% reported to date in kesterite solar cells.

## Results

### Influence of Pd on the selenization process

In previous investigations of the Se vapor evolution in the graphite box, it was found that the Se vapor concentration shows a very significant decrease in the middle and late stages of the selenization[6]. From the perspective of solid-vapor equilibrium at high temperatures, this would induce irreversible escape of Se atom from the CZTSSe crystal surface and GBs, resulting in the formation Se vacancy defect ($V_{Se}$). With the emergence of a large number of $V_{Se}$, the binding of metal atoms on the crystal surface also declines, making the volatile Sn also easy to escape, probably in the form of SnSe vapor, causing the appearance of Sn vacancy defect ($V_{Sn}$)[24,25,35]. Through explorations, we found that the issue of Se and Sn volatilization loss can be effectively addressed by employing Pd incorporated kesterite absorber.

In our approach, $PdCl_2$ was introduced into the precursor solution for the fabrication of Ag-alloyed CZTSSe (ACZTSSe) film, in which Ag was used to promote the film crystallization[36]. For clarity, we refer to the ACZTSSe film with Pd as ACZTSSe-Pd, while the Pd-free ACZTSSe is considered as the control sample. These films were subsequently used for the solar cell fabrication, and the best cell performance was achieved when the Pd/Zn ratio was at 1% (Supplementary Figs. 1 and 2). Through spherical aberration-corrected scanning transmission electron microscopy (STEM) and electron energy loss spectroscopy (EELS) characterization, we found that in the final-state ACZTSSe-Pd film, Pd element is primarily localized along the GB (Fig. 1a, b and

Supplementary Fig. 3). A similar spatial distribution pattern was observed for the Se element in the GB regions (Fig. 1c and Supplementary Fig. 3). It suggests that the introduced Pd was mainly distributed in the GB regions, in the form of $PdSe_x$ compounds such as $PdSe_2$ and PdSe. Besides the GBs, Pd was also detected on the film surfaces by using X-ray photoelectron spectra (XPS), with higher intensity than that in the etched bulk region (Supplementary Fig. 4). This implies that $PdSe_x$ compounds mainly existed both at the GBs and on the film surfaces, forming effective heterogeneous coverings over the ACZTSSe grains. X-ray diffraction (XRD) and Raman investigations indicated that the incorporation of Pd did not alter the size and vibration properties of the ACZTSSe lattice, even when the Pd/Zn ratio was increased to 3% (Supplementary Fig. 5). Even during the high temperature selenization reaction, Pd did not change the relative positions of the XRD peaks of the ACZTSSe phase. Therefore, we speculate that the doping of Pd atoms into the kesterite lattice is negligible. In the ACZTSSe crystallization growth process, Pd was mainly segregated to the GBs and the surfaces, forming $PdSe_x$ coverings. This hypothesis is further supported by density functional theory (DFT) calculations, which shows a high formation energy of >1.0 eV for Pd/M (M = Cu, Zn, or Sn) substitutions in the kesterite lattice (Supplementary Fig. 6).

We further delved into the impact of Pd on the selenization process of ACZTSSe films. We employed X-ray fluorescence (XRF) to quantify the element composition of films sampled at various selenization stages (Fig. 1d and Supplementary Fig. 7). It is evident that the volatilization loss of Sn and Se elements conspicuously and continuously occurred in the control sample when the selenization reaction exceeded 5 minutes. In contrast, the ACZTSSe-Pd sample effectively suppressed element loss, with the Sn/(Cu+Ag+Zn+Sn) ratio remaining nearly constant throughout the entire process. Additionally, a higher and more stable Se/(Cu+Ag+Zn+Sn) ratio was observed during the intermediate selenization process. This suppression of element loss consequently stabilized the electronic structure of Sn. The XPS peak position of Sn in the ACZTSSe-Pd sample remained almost unchanged as selenization proceeded (Fig. 1e, g, and Supplementary Figs. 8 and 9). This is in stark contrast to the control sample, in which the Sn XPS peak continuously shifted towards the lower-energy direction (Fig. 1f, g).

By further fitting, we found that the Sn XPS spectrum of the final-state control sample was composed of two peaks with comparable intensities, locating at 486.9 eV and 486.3 eV, which can be ascribed to Sn(IV) and Sn(II), respectively (Fig. 1g)[37–39]. In contrast, the XPS spectrum of the ACZTSSe-Pd sample was dominated by the higher-energy peak (Fig. 1g). DFT calculations revealed that the appearance of Sn(II) is arisen from $V_{Se}$ defects, which results in lone pair electrons in the 5 s orbit of the nearest neighboring Sn atom (Supplementary Figs. 10 and 11). As such, the continuous shift of Sn XPS spectra during the selenization process is indicative of the ongoing formation of $V_{Se}$, which is a donor defect that would weaken the p-type carriers of the kesterite absorber (Supplementary Fig. 12)[10,35]. This result was confirmed by surface contacting potential difference (CPD) mapping of these films using Kelvin probe force microscopy (KPFM) (Fig. 1h, i and Supplementary Fig. 13). For the control sample, its average CPD evolved from −93 mV to 70 mV as selenization progressed, resulting in an upshift of the Fermi energy level by more than 160 mV. In comparison, the CPD evolution of the ACZTSSe-Pd sample was limited to less than 40 mV (from −129 to −90 mV), demonstrating much more stable surface electrical properties and a more pronounced p-type nature.

### Characterization of ACZTSSe absorbers

We further characterize the impact of Pd on the element distribution in final-state ACZTSSe absorber films using STEM based energy-dispersive X-ray spectroscopy (EDX) analysis. In the case of the

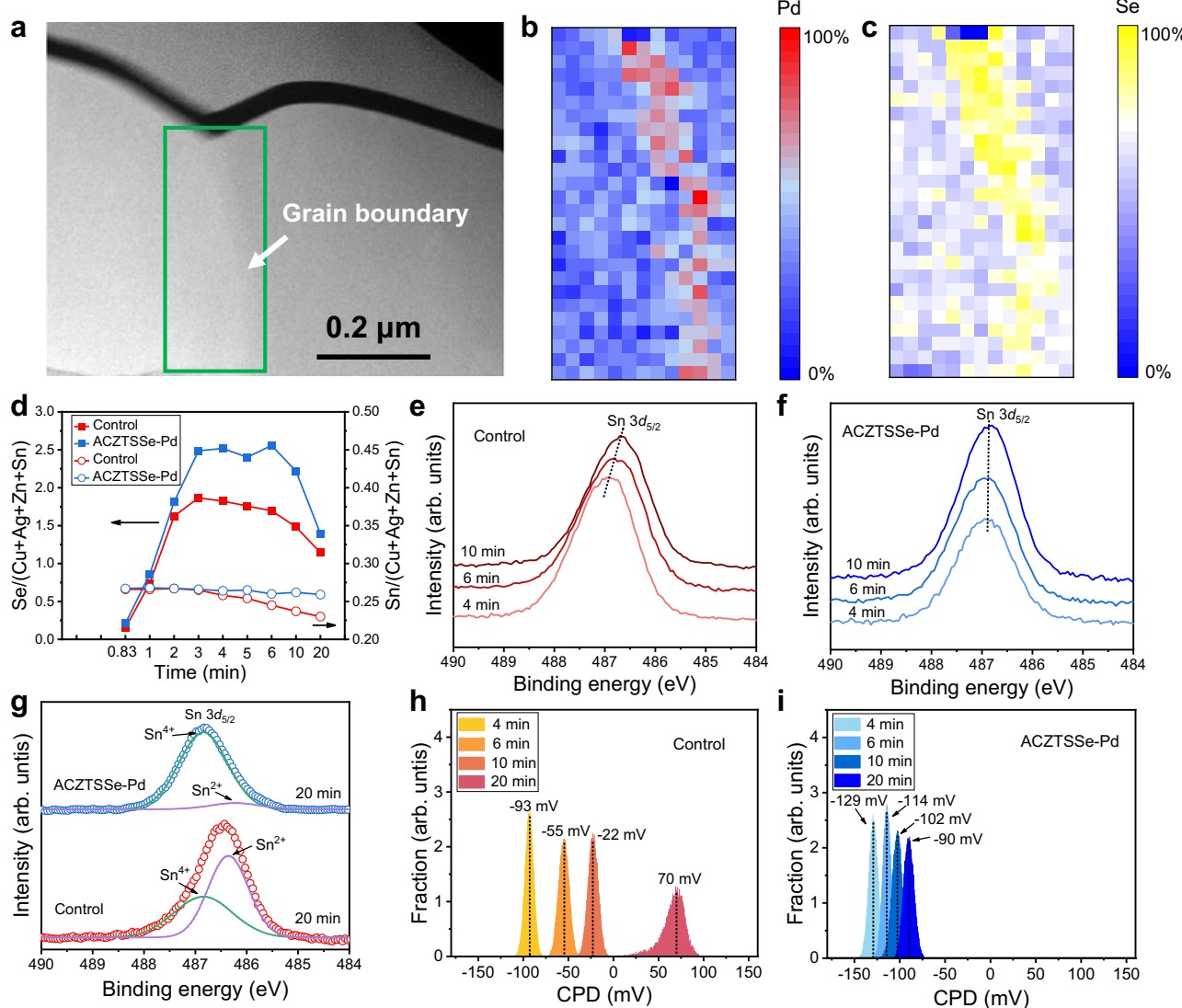

**Fig. 1 | Influence of Pd on the selenization process. a** Cross-sectional HAADF STEM image of the ACZTSSe-Pd film. The green rectangle represents the EELS mapping region. **b, c** EELS mapping image of Pd (M4 and M5 peaks) and Se (L2 and L3 peaks). **d–f** Evolutions of element composition and Sn $3d$ XPS spectra of the films during the selenization process. **g** Gaussian fitting of the XPS spectra of the final-state ACZTSSe films. **h, i** Evolutions of surface CPD of the two samples during the selenization process.

control film (Fig. 2a–d and Supplementary Figs. 14 and 15), a clear deficiency of Sn and Se elements is evident in the GB region, while other elements display a relatively uniform distribution profile across the GBs. This observation indicates that Sn and Se elements in GB regions did indeed experience volatilization loss. In contrast, in the ACZTSSe-Pd film, the Se content in the GB region is slightly higher than that in the GI, and the distribution of Sn is uniform, similar to other elements (Fig. 2e–h and Supplementary Figs. 14 and 15). This suggests that the loss of Sn and Se elements in GB regions has been effectively mitigated. This result was further confirmed on a larger scale using scanning electron microscopy (SEM) (Supplementary Figs. 16 and 17).

CPD mapping results revealed that the suppression of element loss in GBs has significantly altered the energy band bending behaviors of the ACZTSSe absorber film. In the control film, obviously higher CPD was observed in the GB regions compared to the GIs (Fig. 2i), indicating a downward bending of the energy band (Fig. 2k). Under this case, the photo-generated minority electrons would be driven to the defective GB regions and cause carrier loss. The existence of high concentration of donor defects in the GBs is the primary reason for this phenomenon. In contrast, the ACZTSSe-Pd film exhibited an inversion of the energy

band bending (Fig. 2j, l), benefiting from an effective reduction in the $V_{Se}$ donor defect. This upward bending would facilitate the spatial separation of minority electrons, thereby reducing carrier recombination[40–42]. This was confirmed by transient photo-luminescence (PL) measurements (Supplementary Fig. 18), which revealed that the ACZTSSe-Pd film exhibited a significantly prolonged carrier lifetime. Due to the reduced carrier recombination, the ACZTSSe-Pd film also displayed a higher steady-state PL intensity (Supplementary Fig. 19). Moreover, the ACZTSSe-Pd film exhibited a notably smaller PL bathochromic shift relative to its bandgap ($E_g$), more than 20 meV lower than that of the control sample (Fig. 2m). This indicates a reduction in electrostatic potential fluctuations within the absorber film[43–45], which is also evidence of the suppressed defects.

These absorber films were subsequently used to fabricate solar cells following a standard device configuration (Supplementary Fig. 20). We investigated the charge transport and recombination losses of the cells using modulated transient photocurrent/photo-voltage measurements (M-TPC/TPV). It is evident that the ACZTSSe-Pd cell exhibited much faster photocurrent decays at both −1 and 0 V, with the photocurrent rise velocity and peak position remaining

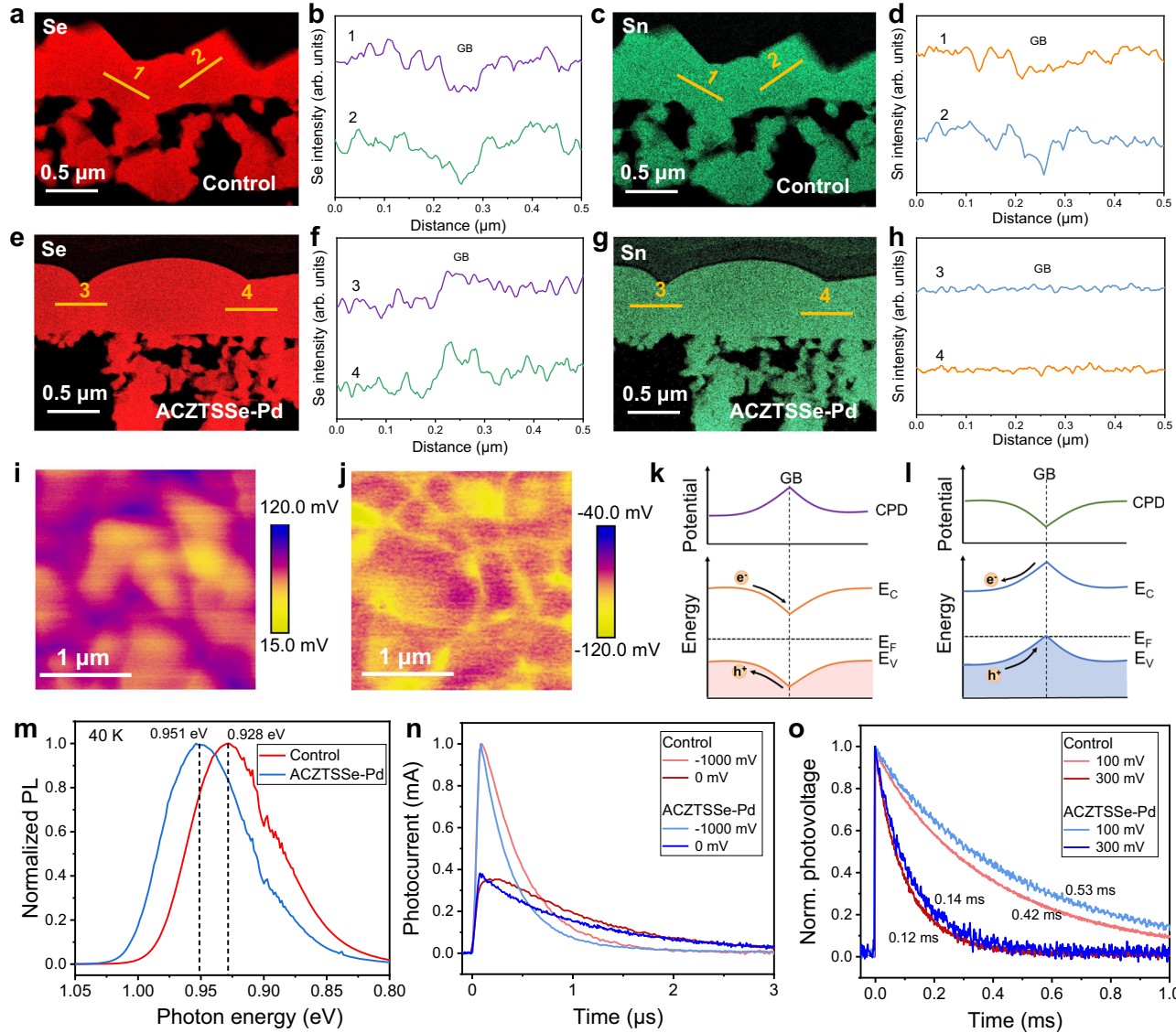

**Fig. 2 | Characterization of the final-state ACZTSSe absorbers. a–h** Cross-sectional STEM-EDX mappings of Se and Sn elements in the films (**a, b** Se, control film, **c, d** Sn, control film, **e, f** Se, ACZTSSe-Pd film, **g, h** Sn, ACZTSSe-Pd film). **i, j** KPFM mapping images of the two absorber films (**i** Control, **j** ACZTSSe-Pd) and (**k, l**) schematic diagram of the energy band bending near the GB regions (**k** Control, **l** ACZTSSe-Pd). "e⁻" represents electron and "h⁺" represents hole. **m** Steady-state PL spectra of the absorber films. **n–o** Modulated transient photo-current and photovoltage dynamics of the cells.

unaffected by the bias voltage (Fig. 2n). In contrast, these dynamics in the control cell were noticeably delayed when the built-in electric field within the cell was weakened by altering the bias voltage (Fig. 2n). These findings demonstrate that the impact of electrostatic potential fluctuations and carrier trapping processes on the charge transport have been effectively reduced in the ACZTSSe-Pd absorber[46,47]. These improvements also suppressed charge recombination, as evidenced by the slower photovoltage decay observed in the cell under different positive bias voltages (Fig. 2o).

## Device performance and characterization

The top-performing cell achieved an impressive power conversion efficiency (PCE) of 14.5%, with a short-circuit current density ($J_{SC}$) of 36.7 mA cm⁻², a $V_{OC}$ of 0.555 V, and a fill factor (FF) of 0.712 (Fig. 3a). In comparison, the control cell exhibited a lower PCE of only 12.8%, with noticeably lower $J_{SC}$ at 35.5 mA cm⁻², $V_{OC}$ at 0.519 V, and FF at 0.695. A statistical analysis of these parameters further highlighted the per-formance disparity between these two cell types (Supplementary Fig. 21). External quantum efficiency (EQE) spectra indicated that the

absorbers in both cells had a similar bandgap of 1.1 eV. The improve-ment in $J_{SC}$, approximately 1 mA cm⁻², primarily originated from EQE enhancement in the wavelength range from 600 to 1080 nm (Sup-plementary Fig. 22). This suggests that photocarriers generated within the bulk absorber were more effectively extracted[4,48], due to reduced charge losses within the GBs. The reduction in charge losses is also responsible for the significant improvements in $V_{OC}$ and FF.

Moreover, the cell has received a certified PCE of 14.3% from an accredited independent testing laboratory, the National PV Industry Measurement and Testing Center (NPVM) (Fig. 3B and Supplementary Fig. 23). The maximum power point tracking (MPPT) of the cell was also conducted in the certification process. When biased at 435.9 mV, the cell gave constant current output of about 32.95 mA cm⁻² for 300 s, achieving a steady-state PCE of approaching 14.4% (Fig. 3c). This achievement stands as the highest result reported to date.

We further quantified the charge loss in the cell based on the modulated electrical transient measurements (Fig. 3d). The primary difference between these two cells lies in the charge extraction effi-ciency ($\eta_{ext}$), which is correlated to the charge loss in the bulk

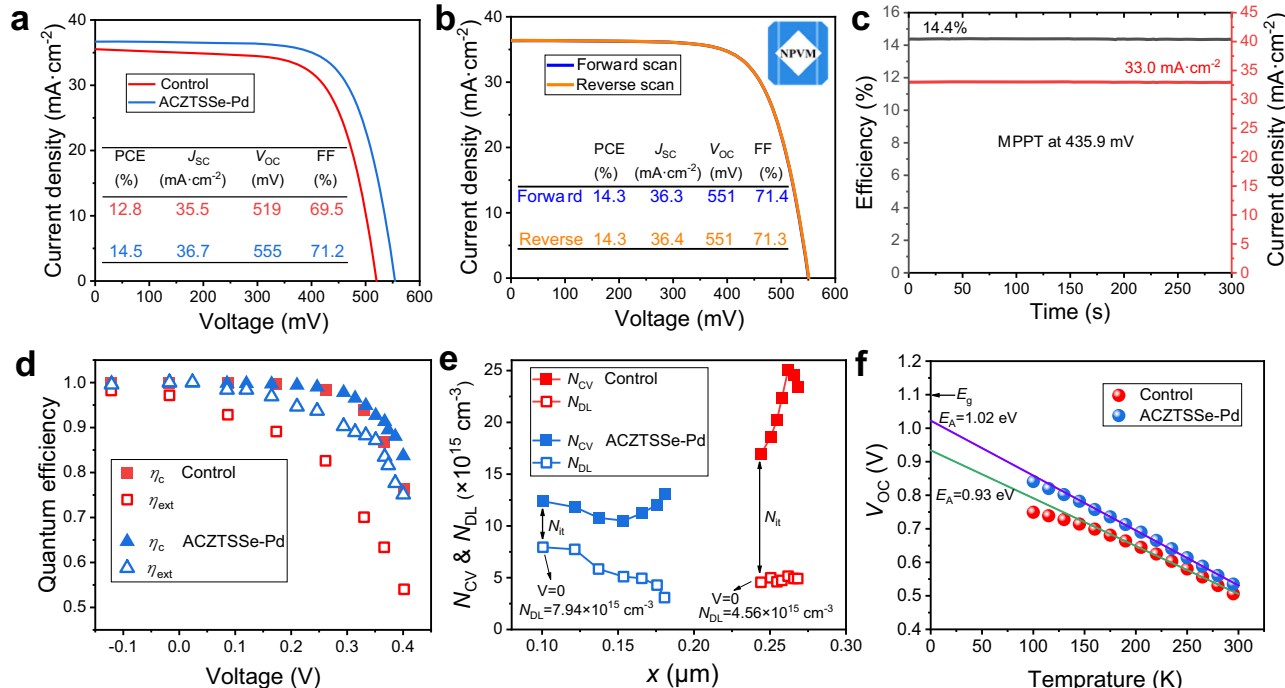

**Fig. 3 | Device characterization. a** Current density-voltage (*J–V*) characteristics of the champion control and ACZTSSe-Pd devices. **b** Certified *J–V* curves of the cell at both forward and reverse scanning directions. **c** Maximum power point tracking of the cell for 300 s. **d** Charge extraction and collection efficiencies of the cells derived from modulated electrical transient measurements. **e** Charge profiles of the cells measured by DLCP and *C–V* at 11 kHz. **f** Temperature-dependent *V*$_{OC}$ of the cells.

absorber. At 0.4 V, the ACZTSSe-Pd solar cell achieved a 1.4-fold enhancement in $\eta_{ext}$. Furthermore, the ACZTSSe-Pd solar cell also exhibited higher charge collection efficiency ($\eta_C$) at voltages exceeding 0.35 V. This indicates that interface defect-induced charge loss in the cell has also been reduced. The findings from the electrical transient analysis are further substantiated by a direct measurement of charge spatial distribution within the cell employing capacitance-voltage (*C-V*) and drive-level capacitance profiling (DLCP) methods (Fig. 3e). Based on the difference in charge density measured by DLCP and *C-V*[49], the volume defect density ($N_{IT}$) near the ACZTSSe/CdS interface region in the ACZTSSe-Pd solar cell is estimated to be $4.4 \times 10^{15}$ cm$^{-3}$, which is approximately one-third of that observed in the control solar cell ($1.2 \times 10^{16}$ cm$^{-3}$). Furthermore, the ACZTSSe-Pd absorber exhibited a higher charge density at 0 V, indicative of improved p-type doping. This aligns well with the CPD mapping results and serves as direct evidence of the suppressed $V_{Se}$ donor defect in the surface region. The enhancement of film surface quality is also evident from temperature-dependent $V_{OC}$ measurements[50]. In the ACZTSSe-Pd device, the activation energy ($E_A$) was determined to be 1.02 eV, closely matching the optical $E_g$ of the absorber (1.1 eV), whereas the control device exhibited a considerably lower $E_A$ of 0.93 eV (Fig. 3f).

**Microscopic mechanism of the Pd-assisted GB engineering**
We proceeded to explore the microscopic mechanisms underlying the defect regulation at GBs after Pd incorporation. To ascertain the form in which PdSe$_x$ compounds existed, we measured XPS spectra of the Pd element throughout the selenization process. In the precursor film, Pd predominantly exhibited a +2-valence state (Supplementary Fig. 24). During the intermediate selenization reaction stages (at 4 minutes and 6 minutes), a proportion of Pd appeared in the +4-valence state (Fig. 4a), indicating the formation of PdSe$_2$, while other Pd ions remained in the +2-valence state, possibly as PdSe. These PdSe$_x$ compounds were also corroborated through XRD and Raman characterization (Supplementary Figs. 25 and 26). Through DFT calculations, we discovered that PdSe$_2$ and PdSe possess higher cohesive

energy than the corresponding SnSe$_x$ compounds (Fig. 4b), signifying superior compound stability and lower volatility. Consequently, from a reaction dynamics perspective, the PdSe$_x$ heterogeneous coverage could mitigate element volatilization from the GBs and surfaces of the ACZTSSe absorber by creating a locally saturated environment. This, in turn, reduces the formation of $V_{Sn}$ and $V_{Se}$ defects. Since this effect accompanied the entire process of crystal growth, the vacancy defects inside the CZTSSe grains should also be reduced accordingly.

Moreover, during the later stages of selenization, XPS results indicated a disappearance of the +4 valent Pd, with Pd predominantly remaining in the +2-valence state (Fig. 4a). This phenomenon was further validated by XRD and Raman spectra (Supplementary Figs. 25 and 26). The valence state evolution of Pd suggests the occurrence of redox reactions. Firstly, through selenization conducted at different temperatures (as described in Supplementary Note 1), we observed that the oxidation of Pd(II) to Pd(IV), leading to the formation of PdSe$_2$, readily occurred even at relatively low temperatures and under low-Se vapor partial pressure (Fig. 4c1 and Supplementary Fig. 27). Secondly, we discovered that PdSe$_2$ exhibits a high oxidation ability when it is mixed with SnSe at elevated temperatures (Supplementary Note 2). XRD analysis indicated that SnSe would undergo oxidations to form SnSe$_2$, while PdSe$_2$ was reduced to form PdSe$_x$ compounds, such as PdSe and Pd$_{17}$Se$_{15}$ (Fig. 4c2), as described by the reaction:

$$PdSe_2 + SnSe \rightarrow PdSe_x + SnSe_2 \qquad (1)$$

The occurrence of this redox reaction was substantiated through DFT calculations, which revealed an enthalpy change of −3.3 kJ/mol at $x = 1$. Furthermore, the notable difference in standard electrode potential ($E^\theta$) between Pd(IV)/Pd(II) ($E^\theta > 1$ V) and Sn(IV)/Sn(II) ($E^\theta < 0.2$ V) provided additional support for this redox reaction[51]. These findings imply that PdSe$_2$ can assist in maintaining Sn in a +4-valence state by supplying Se atoms and capturing excess electrons. In our view, this redox mechanism is also applicable to the mixing system comprising

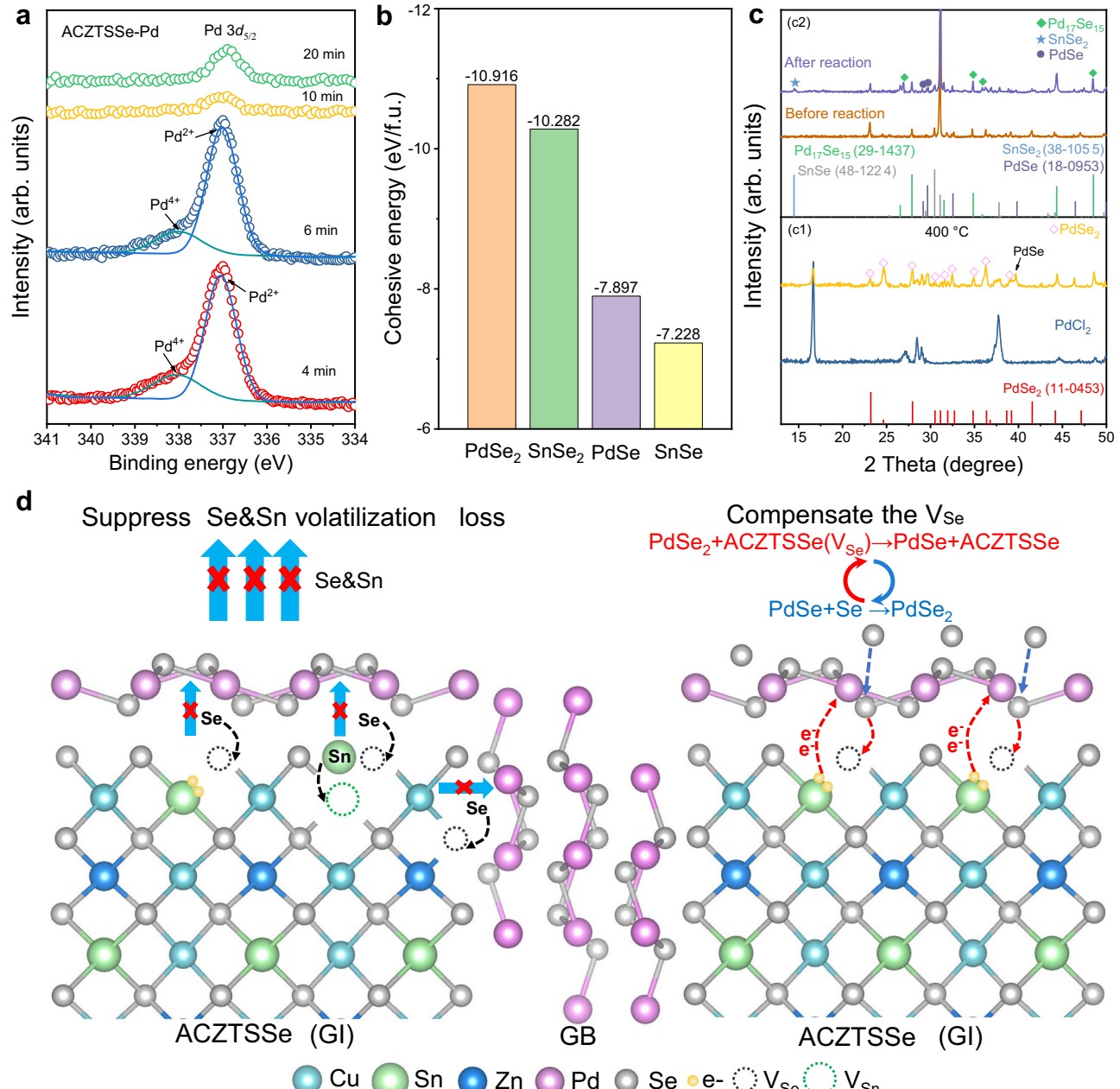

**Fig. 4 | Microscopic mechanism of the Pd-assisted defect regulation in the GB regions. a** XPS spectra of the Pd element during the selenization process. **b** Cohesive energies of PdSe$_2$, SnSe$_2$, PdSe and SnSe. **c** c1 XRD patterns of PdCl$_2$ (bottom) and its selenization product PdSe$_2$ (top); c2 XRD patterns of PdSe$_2$ and SnSe mixture (bottom) and their reaction products (top). **d** Schematic diagram of possible roles that Pd played in the selenization process. Left: preventing element volatilization loss through heterogeneous coverings; right: compensating the V$_{Se}$ defect through Se atom exchange between reaction environment, PdSe$_2$ and ACZTSSe film with Se vacancies.

PdSe$_2$ and ACZTSSe containing low-valence Sn and V$_{Se}$, as follows:

$$PdSe_2 + ACZTSSe(V_{Se}) \rightarrow PdSe + ACZTSSe \qquad (2)$$

This scenario is reasonable because the electron and Se transfer in reaction (2) closely resembles that in the confirmed reaction (1), primarily occurring between Pd(IV) and Sn(II). It is evident that the occurrence of reaction (2) will assist in mitigating the V$_{Se}$ defects that have already formed in the GB regions and on the surfaces of ACZTSSe absorbers. For clarity, we have schematically depicted the microscopic mechanism of Pd-assisted defect suppression in the kesterite absorber in Fig. 4d. First, the PdSe$_x$ compounds formed during the selenization process act as a heterogeneous covering layer of the GBs and the

absorber's surface. This effectively suppresses the Sn and Se element volatilization, preventing the formation of V$_{Sn}$ and V$_{Se}$ defects. Second, Pd(II)/Pd(IV) functions as a redox shuttle, capturing vapor Se from the reaction environment to form PdSe$_2$. Subsequently, PdSe$_2$ can provide Se atoms to the ACZTSSe absorber by being reduced to PdSe, thereby aiding in the elimination of V$_{Se}$ defects. Overall, this redox reaction mechanism offers a promising avenue for the precise regulation of defects in kesterite solar cells, and also holds implications for the GB engineering in other photoelectric devices.

## Discussion

In this study, we focused on the vacancy defects at GBs in kesterite solar cells and have implemented a redox reaction strategy utilizing

palladium (Pd) to suppress these defects. $Pd^{2+}$ was introduced into the ACZTSSe precursor solution, subsequently undergoing reactions to form $PdSe_x$ compounds ($PdSe$ and $PdSe_2$). These compounds can heterogeneously cover the GBs and surfaces, thus suppressing the Sn and Se volatilization loss and the formation of $V_{Sn}$ and $V_{Se}$ defects. Moreover, Pd(II)/Pd(IV) can perform as a redox shuttle, capturing vaporized Se from the reaction environment and subsequently supplying Se atoms to the ACZTSSe absorber, further diminishing the already-existed $V_{Se}$ defects. These effects have effectively reduced defects and additionally enhanced the p-type nature of the ACZTSSe absorber, thus resulting in significantly suppressed charge non-radiative recombination loss in the ACZTSSe solar cell. Consequently, we have achieved the highest certified efficiency of 14.3% reported to date for kesterite solar cells.

Our study here and previously reported works have widely demonstrated the promising role of cation incorporation in enhancing the performance of kesterite solar cells. However, the introduction of a variety of cations also complexes this material system and causes controversies about the material physical mechanisms through which these cations exert their effects, alloying, doping or others. This demands us to better understand the crystallization growth and defect formation processes in the CZTSSe material. In particular, we need to pay more attention to the effects of these incorporated cations on the initial and intermediate states of the CZTSSe material, rather than just on the semiconducting and defective properties of the final-state material. These efforts will help us more effectively determine the origins for current issues of CZTSSe materials and thus guide us more synergistically exploit the positive role of these cations.

## Methods

### Materials
CuCl (99.999%, Alfa), $Zn(CH_3COO)_2$ (99.99%, Aladdin), $PdCl_2$ (59-60% Pd, Innochem), $SnCl_4$ (99.998%, Macklin), $SnCl_4$ (99.998%, Macklin), AgCl (99.5%, Innochem), thiourea (99.99%, Aladdin, recrystallized twice), 2-methoxyethanol (MOE, 99.8%, Aladdin), Se pellets (99.999%, Zhong Nuo Advanced Material), $CdSO_4·8/3H_2O$ (99.99%, Aladdin), and ammonia (25.0-28.0%, Sinopharm Chemical Reagent Co. Ltd.) were used. The chemicals were used directly without further purification.

### Film preparation
ACZTSSe precursor film preparation: The control ACZTSSe precursor solution was prepared by dissolving 7.311 g thiourea, 2.160 g CuCl, 0.345 g AgCl, 3.126 g $Zn(Ac)_2$ and 3.963 g $SnCl_4$ into 30 ml MOE and stirred at 50 °C to obtain a colorless solution. The Pd incorporated precursor solution was prepared by adding Pd contents of 0.5%, 1%, 2%, 3% (Pd/Zn, molar ratio) into the control ACZTSSe precursor solution. All the above steps are carried out in a $N_2$-filled glove box. The obtained precursor solution was spin-coated on Mo substrate at 3000 rpm for 25 s, followed by annealing on a 280 °C hot plate in the air. This coating-annealing process was repeated several times to give a precursor film with ~1.5 μm·thickness. Then, precursor films were placed in a graphite box containing Se particles and selenized in a rapid heating tube furnace. The detailed selenization condition was as followed: the temperature was first raised to 540 °C within 1 min and maintained for 19 min, then reduced naturally to room temperature.

### Device preparation
The ACZTSSe devices were fabricated with a typical structure of Mo/ACZTSSe/CdS/ZnO/ITO/Ni/Al/MgF₂. A 40–50 nm CdS buffer layer was deposited at 70 °C using chemical bath deposition (CBD) method. Specifically, ACZTSSe films were immersed into the aqueous solution (200 ml water) pre-dissolved $CdSO_4$ (~12 mM) and ammonia (~10 ml) in a beaker. Thiourea (~48 mM) was then added into the solution. After the thiourea was completely dissolved, the beaker was immersed into the 70 °C water bath to start the CdS deposition. The CBD was

performed for about 10–11 min to get the desired thinckness. These films were then cleaned by water and dried by $N_2$ for the following window layer deposition. Afterwards, a 20–30 nm ZnO layer and a 180 nm ITO layer were deposited by magnetron sputtering technique. A Ni (40 nm)/Al (2 μm) metal grid were deposited by thermal evaporation to complete the whole device. Finally, a $MgF_2$ layer was thermally evaporated as the anti-reflection coating (ARC).

### Film characterization
Morphologies of the absorber films were measured on Hitachi S4800 scanning electron microscope (SEM). Element mappings were obtained by energy-dispersive spectrometer (EDS, AZtec X-Max 50). The microstructure and elemental distribution were also measured by a JEOL-F200CF scanning transmission electron microscope (STEM) equipped with an EDS system. Electron energy loss spectroscopy (EELS) was obtained using a beam of an electron microscope. XPS measurement was carried out on an ESCALAB 250Xi (Themo Fisher) instrument. X-ray diffraction (XRD) patterns were collected by using an X-ray diffractometer Cu Ka as the radiation source (Empyrean, PANaltcal). Raman spectra were carried out on Raman spectrometer (Lab-RAM HR Evolution, HORIBA) by using 532 nm laser diode as the excitation source. The compositions of films were determined by an energy-dispersive X-ray fluorescence (XRF) spectrometer (EDX-7000, Shimadzu). Atomic force microscope (AFM) and Kelvin probe force microscope (KPFM) images were obtained on Bruker, Multimode 9. For the KPFM, the CPD was measured as the potential difference between the sample and the probe tip, that is $CPD = \phi_{sample} - \phi_{tip} = (W_{tip} - W_{sample})/e$, where $\phi$ is the electric potential, $W$ is the work function and $e$ is the elementary charge. Temperature-dependent steady-state PL and time-resolved PL spectra were obtained on PL spectrometer, FLS 900, Edinburgh Instruments, excited with a picosecond-pulsed diode laser (EPL-445) with the wavelength of 638.2 nm and measured at 730 nm after excitation.

### Device characterization
The current density-voltage (J-V) curves of the solar cells were measured by using Keithley 2400 Source Meter under simulated AM 1.5 sunlight at 100 mW cm⁻² from a solar simulator (Zolix SS150A) calibrated with Si reference cell (calibrated by National Institute of Metrology, China). The voltage was forward scanned from −50 mV to 600 mV with a scanning rate of 90 mV·s⁻¹. The J-V tests were conducted in air at 25 °C, and no preconditioning of the device was performed before the measurement. In the certification in NPVM, the mask area is 0.2694 cm². External quantum efficiency (EQE) was measured by Enlitech QE-R test system using calibrated Si and Ge diodes as references. Modulated transient photocurrent and photovoltage (m-TPC/TPV) measurements were obtained by our lab-made setup, in which the cell was excited by a 532 nm pulse laser (Brio, 10 Hz, 4 ns) and the decay process was recorded by a sub-nanosecond resolved digital oscilloscope (Tektronix, DPO 7104). The C-V profiles, driven level capacity profiles (DLCP) and temperature-dependent J-V were measured with an electrochemical workstation (Versa STAT3, Princeton)

### DFT calculation
All the calculations were performed within the density functional theory (DFT) as implemented in Vienna ab initio simulation package (VASP) code[52,53]. The projector augmented-wave (PAW)[54] pseudopotentials were employed with an energy cutoff of 479 eV. Since $PdSe_2$, $SnSe_2$ and SnSe are two-dimensional (2D) materials with van der Waals (vdW) interactions, we combined with the exchange-correlation functional of Perdew-Burke-Ernzerhof (PBE)[55] and the vdw correction (DFT-D3)[56] in the calculation of cohesive energies. To correctly assess the bandgap and defect energy levels, we employed the hybrid exchange-correlation functional of Heyd-Scuseria-Ernzerhof (HSE06)[57] in the calculation of defect formation energies. A $6 × 6 × 3$ k-mesh[58] was

used to optimize the conventional cell and only the Γ point was considered for the Brillouin zone integration of $2 \times 2 \times 2$ expanded supercell (128 atoms). The atom coordinates were optimized until the residual forces were less than 0.01 eV/ Å. The obtained bandgap is 0.97 eV, which agree well with previous reported results[59–62]. The defect formation energy can be calculated by[63] $\Delta E_f (\alpha, q) = E (\alpha, q) - E (host) + \Sigma n_i \cdot (\mu_i + E_i) + q \cdot (E_F + E_{VBM}) + E_{corr}$, where $E (\alpha, q)$ is the total energy of the supercell with defect $\alpha$ in charge state $q$ and $E (host)$ is the total energy of the pure $Cu_2ZnSnSe_4$ supercell. $n_i$ represents the number of atoms exchanged with supercell. $\mu_i$ and $E_i$ are elemental chemical potential and the total energy of the pure elemental phase, respectively. $E_{VBM}$ is the valence band maximum (VBM) level in the pure supercell and $E_F$ is the Fermi energy referenced to the VBM level. $E_{corr}$ includes the correction of formation energy with the scheme proposed by Freysoldt[64]. For the chemical potentials used in the defect formation calculation, they were selected from the calculated phase diagram of CZTSe, including typical endpoints of the Cu-Zn-Sn 3-dimensional stable phase region and some representative points at different Cu chemical potentials.

## Reporting summary

Further information on research design is available in the Nature Portfolio Reporting Summary linked to this article.

## Data availability

The photoelectric characterization and device performance data generated in this study are provided in the Supplementary Information/Source Data file. Source data are provided with this paper.

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

## Acknowledgements

This work was supported by the National Natural Science Foundation of China (Grant nos. U2002216 (Q.M.), 52222212 (J.S.), 52172261 (Y.L.), 52227803 (Q.M.), 51972332 (H.W.), 12174060 (S.C.) and 12334005 (S.C.)). J.S. also gratefully acknowledges the support from the Youth Innovation Promotion Association of the Chinese Academy of Sciences (2022006).

## Author contributions

Jinlin Wang, Jiangjian Shi, Dongmei Li, and Qingbo Meng conceived the idea and designed the experiments. Jinlin Wang and Jiangjian Shi did the experiments and the data analysis. Licheng Lou, Jiazheng Zhou, and Xiao Xu supported CZTSSe solar cell fabrication. Fanqi Meng and Kang Yin performed STEM and DLCP measurements. Huijue Wu and Yanhong Luo supported M-TPC/TPV characterization and discussions. Jinlin Wang, Jiangjian Shi, Dongmei Li, and Qingbo Meng participated in writing the manuscript. Shanshan Wang and Shiyou Chen provided DFT results and valuable discussion of the work.

## Competing interests

The authors declare no competing interests.
