## [Peer Review File · Nature Communications]

Pd(II)/Pd(IV) redox shuttle to suppress vacancy defects at grain boundaries for efficient kesterite solar cellsREVIEWER COMMENTS

Reviewer #1 (Remarks to the Author):

This manuscript presented by Wang et al focused on eliminating the vacancy defects of Sn and Se in the grain boundary and surface regions of Kesterite CZTSSe polycrystalline films. It was found that these vacancy defects, especially the Se vacancy, are primarily formed in the later stage of the CZTSSe selenization process due to surface elemental volatilization loss. The Se deficiency in the graphite box in the later-selenization stage is an important reason for this phenomenon. The author proposed a Pd assisted kinetic strategy to suppress the elemental volatilization loss. It was found that Pd could segregate into the GBs of the film and form PdSex compounds. The coverage of PdSex on the film could thus kinetically suppress the Se and Sn loss. They also proposed that the redox reaction between Pd(II) and Pd(IV) through Se capturing and releasing could eliminate the already-formed Se vacancy in the film GBs and surface. Finally, the Pd strategy realized an extraordinarily high efficiency of 14.5% (14.3% certified). This is the most recent and exciting result of this field and represents that CZTSSe solar cells are about to enter the era of 14%.

Another point impressed me in this work is that the author is trying to understand the defect formation from the perspective of practical selenization reaction process. This effort allows researchers to more directly understand the physical process of defect formation, so as to develop corresponding defect regulation methods from the perspective of controlling the reaction process.

Overall, this work is very valuable to this field and thus is acceptable. Nonetheless, some issues still need be addressed to make this manuscript clearer to the readers.

1. A detailed discussion about the physics process of the vacancy defect formation should be given to make the reader more easily follow the author's idea.
2. Since, the Sn and Se volatilization loss is mainly due to the element deficiency in the graphite box in the later-selenization stage, have the author tried adding more Se or adopting other more direct strategy to control the Se in the selenization process?
3. Have you characterized Pd on the film surface, like figure 1a-c? In this work, the author discussed both GB and surface while did not clearly distinguish their difference. Do they have the same issue, or can they be equally treated?
4. Why the STEM morphology in Figure 2A is significantly different from that in 2E? The samples were from selected from different batches?
5. What is the measurement frequency of the CV and DLCP? Does this frequency contain the defect information?
6. What is the physics meaning of the calculated cohesive energy? Is it related to the elemental volatilization?

7. Other phases such as Pd₁₇Se₁₅ also existed in the XRD in Figure 4c. Does this phase also exist in the Pd-ACZTSSe film in the selenization process?
8. Figure 4d is not clear enough to understand. Please improve it. How is the electron moved between different materials?
9. Ag was also used in the film deposition. Can the author add a description of its role?
10. In the DFT calculation section, the author should avoid using formula. Directly using word text and symbols is better.
11. What is current density integration of the EQE spectra in supplementary Fig. 20?

Reviewer #2 (Remarks to the Author):

This manuscript by Meng and colleagues proposes that vacancy defects caused by the inevitable and irreversible volatilization loss of Se or Sn elements at GBs of the CZTSSe film are more important origins for the charge loss. Authors start from the perspective of controlling the volatilization kinetics and develop a local reaction strategy leveraging Pd to eliminate these vacancy defects. This approach is innovative and the reported total-area efficiency of 14.5% is a significant achievement within the kesterite solar cell research community. Therefore, I recommend it to be published after the minor revision. Also, there are some questions to be addressed below:

1. The manuscript only shows the device performance of control sample (without Pd) and ACZTSSe-Pd sample (1% Pd). So what is the device performance under other Pd concentrations, and whether the introduction of Pd has any influence on the morphology of the ACZTSSe absorber?
2. Does the selenization process result in any loss of Pd, despite the presence of Pd in the final ACZTSSe-Pd film as characterized by XPS in this manuscript?
3. What is the existence form of the residual Pd in the ACZTSSe-Pd film? If it exists in the form of a secondary phase, then what is the reason for the absence of Pd-related secondary phase peaks in the XRD patterns?
4. Although DFT shows that Pd prefers to exist at the grain boundaries, and experimental results also detect that Pd is ultimately located at the grain boundaries, does Pd also hinder the volatilization of elements inside particles during the selenization process? Please provide a clear explanation in the revised manuscript.
5. In other literatures (*Adv. Mater.*, 2020, 32, 2000121; *Nat. Energy*, 2018, 3, 764-772) related Kesterite solar cells, the carrier lifetime measured by TRPL is only a few ns, while the carrier lifetime obtained by TRPL in this article is on the order of hundreds of nanoseconds. Please give a reasonable explanation why there is such a big difference in the measured carrier lifetimes?

6. The integrated current density curve should be given in the EQE plot in Supplementary Figure 20. Pd and alkali metal doping share similarities in that both tend to enrich at grain boundaries. However, while alkali metal doping in CZTSSe also passivates defects and enhances the material's carrier concentration and Voc, the device's Jsc is usually reduced. Why does Pd doping in CZTSSe lead to an improvement in the device's Jsc?

7. At around 300°C, Pd²⁺ ions easily react with O₂ to form PdOx. Has PdOx (containing Pd²⁺ and Pd⁴⁺ ions) already been produced during the precursor film preparation process and then transformed into PdSex during the selenization process?

8. During the high-temperature selenization process, Pd elements exhibit a higher diffusion rate in the absorber layer. Please explain why do Pd elements aggregate on the surface of the absorber layer? Additionally, Pd elements are mainly distributed at grain boundaries. In the lower layer of the absorber layer with smaller grains, there are more grain boundaries, theoretically suggesting a higher content of Pd elements.

9. Does the reduction carrier recombination losses at heterojunction interfaces relate to an increase in the bandgap at the absorber layer surface or the easier promotion of carrier transport due to band bending at the absorber layer surface? Would a single gradient distribution of Pd elements alter the original band structure of the absorber layer?

10. ACZTSSe-Pd has achieved a high certification efficiency. The absorber layer material has evolved from the quinary CZTSSe to the septenary ACZTSSe-Pd, making bulk defects already very complex. Please properly propose a future research direction related of cationic doped CZTSSe for better device performance.

Reviewer #3 (Remarks to the Author):

The authors present a comprehensive study about the effects of mixing a small amount of Pd into the precursor solution for solution-processed Cu₂ZnSn(S, Se)₄ (CZTSSe) kesterite solar cells. From a variety of characterization methods, it is concluded that PdSe and PdSe₂ form in the grain boundaries and on the surface of the kesterite film. The main effect of the presence of Pd is concluded to be avoiding the loss of Sn and Se during the selenization treatment, avoiding thereby donor-type selenium vacancy formation, which raises p-type concentration and reduces donor-type grain boundary defects. The introduction of the Pd additive and the developed understanding of its function present a significant advance in the kesterite field, which is additionally demonstrated by the achieved device efficiencies, which surpass previous best performances.

The work is well presented, and the manuscript is straight forward to read. Nevertheless, some shortcomings are presented, which should be addressed prior to publication.

1. Introduction: secondary phases are a major problem in the synthesis process of kesterites. The introduction mentions this topic only marginally and as a relevant issue at grain boundaries (GBs). While the presented study focuses on the effect of GBs, it should be clearly presented that the GBs are not the only reason for the low efficiency values obtained so far. Thus, a brief introduction to the impact of secondary phases on the solar cell performance should be included, to put the main aspect of the GB effects into perspective.
2. The abstract states 14.5% efficiency, while the last paragraph of the introduction states 14.3% efficiency. I suppose this is because one is the lab measurement and the second is the certified value. Nevertheless, it would be preferable to be consistent in the reporting of efficiency values.
3. XPS data (Suppl. Fig. 2) is shown to support the presence of PdSe₂ on the surface. The XPS spectra shown only indicate the Pd 3d(5/2) peak, and the peak on the surface appears to be slightly asymmetric. Can the authors make any conclusion on the binding state from these data? Is the Pd peak shifted corresponding to PdSe₂ bonds? Are reference measurements of a PdSe₂ compound available to substantiate the conclusion toward a PdSe₂ compound?
4. Lines 105-109: With the addition of low ($\leq 3\%$) Pd concentrations, Pd doping would likely not be observed in XRD. Thus, the conclusion that Pd is not present in grains and only segregates to GBs is not sufficiently substantiated with the shown absence of an XRD shift.
5. Supplementary Fig. 4: why were the stated chemical potentials selected? What is their significance?
6. Line 112: It is claimed that the formation energy for Pd substitution in specific sites in the kesterite lattice is larger than 1 eV. What is the formation energy for a Pd substitution in the GB? How does it compare?
7. I would like to encourage the authors to rethink the use of colors in Fig. 1. Some of the panels consistently use blue and red for ACZTSSe-Pd and control samples (as done also in other Figures), which is very helpful to the reader. Other panels (such as (d), (d) and (f) are not consistent with this choice. Please consider using red and blue tones in (e) and (f), similar to (h) and (i). Hopefully also (d) can be changed to be consistent in the use of colors.
8. Do the data in Supplementary Fig. 7 correspond to the final film after 20 minutes selenization time? It would be helpful to state this specifically in the caption.
9. Is the data shown in Supplementary Fig. 10 from DFT or experimental data? This should be clearly stated in the caption.
10. Regarding the KPFM data shown in Fig. 1 (H) and (I), it is essential to know how the measurements were done and how the CPD is defined. Is it work function of tip minus that of sample, or the opposite? Only with this information, the conclusion toward stronger p-type of the Pd sample can be made. This information should be clearly stated in the experimental section. Furthermore, comparing CPD values from sample to sample requires the use of the same tip without any tip modification (e.g. due to accidental tip crashes). Was this assured in the presented experiments?
11. Supplementary Fig. 11: The authors do not comment on the clear contrast at the GBs for control and ACZTSSe-Pd samples. For the latter, this contrast is very clear, while it is weaker for the control sample. Interestingly, the contrast at the GBs seems to invert for both samples after the 20 min selenization. In

the main manuscript in Fig. 2, only the 20 min images are shown and discussed. Since the contrast inverts for these images, a proper analysis and discussion needs to be included to why this might happen and why the GBs at earlier stages in the selenization process show the opposite band bending.

Furthermore, regarding Fig. 2, the figure caption or the images (I) through (L) should indicate which sample is shown. Additionally, the caption indicates that (K) and (L) correspond to KPFM images, while they are obviously schematic drawings of the potential profile at the GBs. The caption should be corrected accordingly.

12. Supplementary Fig. 14: It is not really clear from the images shown that there is a difference between the control sample and the ACZTSSe-Pd sample. What specific features do the white arrows point to? It is not clear how they are different between control and Pd sample.

13. For the solar cell fabrication, was the PdSe₂ which is observed at the absorber surface removed in any way? PdSe₂ is reported to be a low bandgap semiconductor (for more than single layer PdSe₂), which should be expected to be detrimental for PV performance, if present at the pn-junction.

14. More details about the CdS recipe should be provided.

15. Some specific corrections:

- o Line 96: I suppose that it should state ACZTSSe-Pd, as the film shows the presence of Pd in the EELS data. Same in lines

- o Line 107/108: The use of “neither” requires a “nor”. Is this sentence incomplete?

Reply to reviewer 1

Firstly, we would like to show our sincere appreciation to your kind and professional comments, which have helped us to revise and improve this manuscript. In the following, we will reply to the raised questions, suggestion or comments one by one.

Comment 1: A detailed discussion about the physics process of the vacancy defect formation should be given to make the reader more easily follow the author's idea.

Reply: Thanks very much for this comment. The appearance of the vacancy defect is mainly caused by the Se vapor deficiency in middle and late stage of the selenization process and the solid-vapor equilibrium between the CZTSSe film and the Se vapor environment.

In our previous investigations, we have traced the Se concentration evolution in the selenization process (Nat. Energy, 2023, 8, 526-535). As shown in the above figure (left panel), in the middle and late stage of the selenization, the Se concentration decreases very significantly due to the Se leakage from the graphite box. In this case, the Se vapor in the reaction environment could be not sufficient to sustain the stability of the CZTSSe material at high temperatures. The right panel of the above figure shows that a relatively high Se vapor pressure is needed for forming CZTSSe phase. That is, driven by the solid-vapor equilibrium, Se in the CZTSSe film will volatilize into the external environment. Obviously, this phenomenon firstly occurs in the surface region of the film (including grain top surface and grain boundaries) through the escape of Se atom.

This will result in the formation of Se vacancy. With the emergence of a large number of Se vacancies, the binding of metal atoms on the crystal surface also declines, making the volatile Sn also easy to escape, probably in the form of SnSe vapor, causing the appearance of Sn vacancy.

We believe this is the physics mechanism for the vacancy defect formation in the CZTSSe absorber, and for clarity, we have added a description in the manuscript.

“In previous investigations of the Se vapor evolution in the graphite box, it was found that the Se vapor concentration shows a very significant decrease in the middle and late stages of the selenization process. From the perspective of solid-vapor equilibrium at high temperatures, this would induce irreversible escape of Se atom from the CZTSSe crystal surface and GBs, resulting in the formation Se vacancy defect (V_{Se}). With the emergence of a large number of V_{Se} , the binding of metal atoms on the crystal surface also declines, making the volatile Sn also easy to escape, probably in the form of SnSe vapor, causing the appearance of Sn vacancy defect (V_{Sn}).”

Comment 2: Since, the Sn and Se volatilization loss is mainly due to the element deficiency in the graphite box in the later-selenization stage, have the author tried adding more Se or adopting other more direct strategy to control the Se in the selenization process?

Reply: Thanks very much for this comment. Indeed, as you suggested, we had tried to add more Se in the reaction chamber (i.e. graphite box); however, no obvious effect had been obtained. There are several possible reasons. Firstly, the Se leakage and Se concentration evolution is a kinetics process. At high temperatures, more Se source is usually accompanied with more significant Se leakage from the graphite box. Secondly, Se concentration has an obvious influence on the initial selenization reaction pathway. When the initial Se concentration is significantly changed, the reaction kinetics in the film will also be influenced, which may not be beneficial for the final cell performance. Briefly, we have optimized the amount of the Se source in the graphite box, and it was found that just by changing the Se amount the cell performance cannot be obviously improved. Regarding more direct strategy, we previously have explored a two-zone

selenization process (Nature Communications 2023, 14, 6650). In this strategy, liquid Se was firstly introduced onto the film surface to facilitate phase evolution and crystallization. However, in the late stage, we need to use a low Se environment to remove the excessive Se from the bottom region of the film. That is, Se volatilization still occurred in this strategy.

Overall, as suggested, we have explored many other strategies to solve the vacancy issues; however, they all had less cell performance than that obtained by the Pd incorporation as reported in this work. In the future, we will continue to explore more possible pathways to more effectively control the selenization reaction and defect formation process.

Comment 3: Have you characterized Pd on the film surface, like figure 1a-c? In this work, the author discussed both GB and surface while did not clearly distinguish their difference. Do they have the same issue, or can they be equally treated?

Reply: Thanks very much for this comment. In our STEM investigations, we have not measured the surface region. This is because through XPS we have firstly confirmed the existence of Pd element on the film surface, and thus in the STEM measurement, we focused more on the grain boundary. For the surface characterization, XPS is more convenient than the STEM because it does not need precise film preparation process. In our study, we have used XPS to trace the evolution of film surface during the selenization process.

Indeed, as you said, in this work we treated the grain surface and boundary equally. Grain surface and boundary are both terminals of a grain, which usually contain large amount of defect and lattice distortions. In most cases of CZTSSe selenization, Se will

adsorb into the film and participate in assisting the mass transport and crystal growth. That is, Se will also exist in the grain boundary region of the film. As such, the grain surface and boundary could face similar chemical environments. In addition, both of them are important path in the Se volatilization loss and thus face the similar solid-vapor equilibrium issue. As such, when we study the element volatilization loss and vacancy defect in this work, these two regions can be treated equally.

Comment 4: Why the STEM morphology in Figure 2A is significantly different from that in 2E? The samples were selected from different batches?

Reply: Thanks very much for this comment. These two samples were from the same fabrication batch while their STEM measurements were performed in different batches. Two possible reasons could explain the difference in the STEM image. Firstly, from the AFM and SEM images, we can see that these two samples have a little different morphology. The ACZTSSe-Pd sample has a little larger grain size. Secondly, in the STEM sample preparation process, the surface of the control sample was less effectively protected by C/Pt, which may lead to the damage of the film surface, making it look rougher than that of the target sample. Nonetheless, since we here focus more on the grain boundary, the surface damage should have little influence on our study.

Comment 5: What is the measurement frequency of the CV and DLCP? Does this frequency contain the defect information?

Reply: Thanks very much for this comment. These two measurements were performed with an AC frequency of 11 kHz. We have added this information in the caption of figure 3. According to our previous result (Nature Communications 2023, 14, 6650), defects had capacitance response in this intermediate frequency region.

Comment 6: What is the physics meaning of the calculated cohesive energy? Is it related to the elemental volatilization?

Reply: Thanks very much for this comment. Cohesive energy refers to the energy released when individual atoms or molecules in a crystal are bonded to each other

through chemical bonds to form a crystal. It represents the stability and structural strength of the crystal. This energy was calculated by comparing the total energy of the metal selenide crystal and its atom components. As such, cohesive energy is an appropriate parameter to reflect the thermodynamic properties of element volatilization.

Comment 7: Other phases such as $\text{Pd}_{17}\text{Se}_{15}$ also existed in the XRD in Figure 4c. Does this phase also exist in the Pd-ACZTSSe film in the selenization process?

Reply: Thanks very much for this comment. We totally agree with you that other Pd-Se phases such as $\text{Pd}_{17}\text{Se}_{15}$ could also exist in the Pd-ACZTSSe sample in the selenization process. However, since the amount of Pd in the sample is very low, no obvious $\text{Pd}_{17}\text{Se}_{15}$ signatures can be observed from the XRD and Raman results, as shown in the below figure. For XPS, $\text{Pd}_{17}\text{Se}_{15}$ and PdSe have similar chemical valence and thus can hardly be distinguished. Overall, although Pd-Se could have several types of compositions and structures, their chemical valences are mainly in the form of +4 or +2. As such, we have not completely distinguished these phases in this work, and in some places, we used PdSe_x to denote the Pd-Se compounds.

Comment 8: Figure 4d is not clear enough to understand. Please improve it. How is the electron moved between different materials?

Reply: Thanks very much for this comment. We have improved Figure 4d, as shown in the below figure.

In this revision schematic diagram, the possible vacancies are depicted with dashed circles to more clearly show the elemental volatilization loss. In addition, the redox reaction has also been added in the right panel.

Regarding the electron movement, in our opinion, it is realized through redox reaction and the transfer of Se atoms between different compounds.

Comment 9: Ag was also used in the film deposition. Can the author add a description of its role?

Reply: Thanks very much for this comment. Ag alloying has been widely used to promote the crystallization growth of CZTSSe film and it was also found to be able to reduce the defect. Its effect has been previously studied in several literatures such as 10.1002/adfm.202101927; arXiv:2306.14629; 10.1002/advs.202302869.

We have added a sentence regarding the Ag in the manuscript as “**In our approach, PdCl₂ was introduced into the precursor solution for the fabrication of Ag-alloyed CZTSSe (ACZTSSe) film, in which Ag was used to promote the film crystallization.**”

Comment 10: In the DFT calculation section, the author should avoid using formula. Directly using word text and symbols is better.

Reply: Thanks very much for this comment. We have changed the formulas to text and symbols.

Comment 11: What is current density integration of the EQE spectra in supplementary Fig. 20?

Reply: Thanks very much for this comment. We have added the integrated current density to Supplementary Fig. 22, as shown below. The integrated J_{SC} of the target cell is about 38.5 mA cm^{-2} , which agrees well with the total-area J_{SC} (36.7 mA cm^{-2}) of the cell shown in Figure 3.

Reply to reviewer 2

Firstly, we would like to show our sincere appreciation to your kind and professional comments, which have helped us to revise and improve this manuscript. In the following, we will reply to the raised questions, suggestion or comments one by one.

Comment 1: The manuscript only shows the device performance of control sample (without Pd) and ACZTSSe-Pd sample (1% Pd). So what is the device performance under other Pd concentrations, and whether the introduction of Pd has any influence on the morphology of the ACZTSSe absorber?

Reply: Thanks very much for this comment. Statistical performance parameters of the device at varying concentrations of Pd have been added in supplementary Fig. 2. It can be seen that the J_{SC} , FF and V_{OC} exhibit a trend of increasing followed by a subsequent decrease with the increase of Pd concentration. 1% Pd shows the highest J_{SC} , FF and V_{OC} , culminating in an average efficiency of approximately 14.1%. We have also added the morphological analysis of the CZTSSe absorber at various Pd concentrations characterized using SEM into the supplementary information (supplementary Fig. 1). As can be seen, the incorporation of Pd has not significantly changed the morphology of the film. Nonetheless, the 1% Pd sample seems have a little smoother surface. According to our experience, this small difference in the morphology would not have obvious influence on the cell performance.

Supplementary Figure 1. The top-view and cross-sectional SEM images of ACZTSSe absorber with different Pd concentrations:(a) 0% Pd, (b) 0.05% Pd, (c)1% Pd, (d) 2% Pd, (e)3% Pd.

Supplementary Figure 2. Statistics performance parameters analysis of the device with different Pd concentrations: (A) J_{sc} , (B) V_{oc} , (C) FF, (D) PCE. Each box contains 18 solar cells.

Comment 2: Does the selenization process result in any loss of Pd, despite the presence of Pd in the final ACZTSSe-Pd film as characterized by XPS in this manuscript?

Reply: Thanks very much for this comment. According to the XPS results shown in Figure 4A, we think the loss of Pd also occurred in the selenization process. As can be seen in Figure 4A, the XPS peak intensity of Pd in the 20 min-selenization sample is much lower than that selenized for 4 and 6 min. This obvious reduction in the XPS intensity could be an indication for the reduced amount of Pd on the film surface. Nonetheless, despite the Pd can also volatilize in the high temperature selenization process, its heterogeneous covering on the film surface and grain boundaries can still suppress the loss of Sn and Se.

Comment 3: What is the existence form of the residual Pd in the ACZTSSe-Pd film? If it exists in the form of a secondary phase, then what is the reason for the absence of Pd-related secondary phase peaks in the XRD patterns?

Reply: Thanks very much for this comment. As depicted in Figure 4A of the manuscript, the XPS peak for Pd 3d in the final-stage ACZTSSe film (20 min) exhibits relative symmetry and a lower binding energy, suggesting that the residual Pd predominantly

exists in a reduced valence state, probably +2. We speculate that this residual Pd is likely situated at grain boundaries or on the surface in the form of lower-valence Pd compounds, such as PdSe or Pd₁₇Se₁₅. The potential reasons for the non-detection of these compounds by XRD are as follows: (i) the concentration of these compounds is below the detection limit of XRD; (ii) these compounds could be amorphous or possess low crystallinity. According to the above analysis, several facts we can confirm, that is, (1) Pd still exist in the film; (2) Pd has not doped in the ACZTSSe lattice; (3) Pd existed in the grain surface and grain boundary region; (4) Pd is in a low-valence state.

Comment 4: Although DFT shows that Pd prefers to exist at the grain boundaries, and experimental results also detect that Pd is ultimately located at the grain boundaries, does Pd also hinder the volatilization of elements inside particles during the selenization process? Please provide a clear explanation in the revised manuscript.

Reply: Thanks very much for this comment. This is an interesting point when we consider the crystallization growth process of CZTSSe.

Here, we use the above figure to schematically show the crystallization growth process of CZTSSe film. Due to the negligible doping of Pd into the CZTSSe lattice, it would be spontaneously segregated to cover the CZTSSe grains always from the nucleation to the final stage. The growth of the CZTSSe crystals is realized by the migration of GBs and is driven by consuming the elements from the bottom region. In this case, the effect of suppressing element volatilization loss exists during the entire selenization process. As such, we think that the covering of Pd could also reduce the element vacancy within the CZTSSe grains. As suggested, we have added a sentence about this case as “**Since this effect accompanied the entire process of crystal growth, the vacancy defects inside the CZTSSe grains should also be reduced accordingly.**”

Comment 5: In other literatures (Adv. Mater., 2020, 32, 2000121; Nat. Energy, 2018,3, 764-772) related Kesterite solar cells, the carrier lifetime measured by TRPL is only a few ns, while the carrier lifetime obtained by TRPL in this article is on the order of hundreds of nanoseconds. Please give a reasonable explanation why there is such a big difference in the measured carrier lifetimes?

Reply: Thanks very much for this comment. The carrier recombination dynamics and the PL decay lifetime of semiconductor materials is significantly influenced by temperature. It is generally observed that as temperature rises, carrier lifetime tends to diminish due to the enhanced carrier non-radiative transition dynamics. In the mentioned literatures (Adv. Mater., 2020, 32, 2000121; Nat. Energy, 2018,3, 764-772), the TRPL was mainly measured at room temperature, which only shows lifetime of several nanosecond. In contrast, our TRPL was measured at low temperatures (50 K). At this temperature, the carrier should have a much longer non-radiative lifetime. In some literatures (Appl. Phys. Lett., 2013, 103, 103506), carrier lifetime reaching 10 μ s was also measured at lower temperature (4 K). We know room temperature is the practical operation condition of the cell. However, the PL signal of CZTSSe materials can hardly be measured out at this temperature, especially using the commercial low-excitation (~ 10 nJ cm⁻²) PL instrument. Although the PL can be measured using much higher excitation, the very high carrier injection can also change the intrinsic carrier recombination process. In addition, the lifetime of several nanosecond has approached the instrument response function and thus is usually difficult to be accurately measured. This is why we measured TRPL at relatively low temperatures, which on one hand can avoid the use of very high injection and on the other hand can ensure the lifetime be accurately measured and clearly compared.

Comment 6: The integrated current density curve should be given in the EQE plot in Supplementary Figure 20. Pd and alkali metal doping share similarities in that both tend to enrich at grain boundaries. However, while alkali metal doping in CZTSSe also passivates defects and enhances the material's carrier concentration and Voc, the device's Jsc is usually reduced. Why does Pd doping in CZTSSe lead to an improvement

in the device's J_{sc} ?

Reply: Thanks very much for this comment. As you suggested, the EQE integration has been added in Supplementary Figure 22. The integrated J_{sc} of the target cell is about 38.5 mA cm^{-2} , agreeing well with the I-V result shown in Figure 3.

Regarding the comparison between Pd and the widely used alkali metal doping, in our opinion, they differ in several aspects. Firstly, Li can dope into the CZTSSe lattice and a little increase the bandgap of the absorber, which is an important reason for the reduced J_{sc} . Comparatively, Pd cannot dope into the lattice and thus does not change the bandgap and light absorption properties of the CZTSSe. Its assisted reduction in the defect at grain boundary and surface regions will improve the charge transport and collection and thus enhance the J_{sc} . Secondly, alkali metal selenides such as Li_xSe and Na_xSe are usually considered as flux assistance for the growth of CZTSSe crystals. This implies that the alkali metal elements also have high diffusion and volatilization ability, thus can hardly suppress the element loss. Thirdly, the passivation role of alkali metal elements mainly lies in breaking up the Se-Se dimer, whose mechanism is different from that of Pd reported in this work. Fourthly, the increase in hole density caused by the alkali metal doping, particularly the Na element, is mainly realized by introducing more shallow acceptors such as V_{Cu} . While for the Pd, the increase in carrier concentration is mainly realized by reducing the V_{Se} donor defects and its induced charge compensation effect. As such, their physics mechanism is different.

Overall, both alkali metal doping and Pd incorporation can enhance the cell

performance, although adopting different mechanism. To achieve higher performance, we need to synergistically exploit their positive effects.

Comment 7: At around 300 °C, Pd²⁺ ions easily react with O₂ to form PdO_x. Has PdO_x (containing Pd²⁺ and Pd⁴⁺ ions) already been produced during the precursor film preparation process and then transformed into PdSe_x during the selenization process?

Reply: Thanks very much for this comment. This is an interesting point and honestly, we have not considered it before. Here we look back at the Raman spectra of the control and Pd samples in the initial selenization stage. If a considerable amount of Pd has transformed into Pd oxides, these phases should still exist in the initial selenization stage. However, as shown in the below figure, no Raman signal corresponding to PdO can be observed in the Pd samples, implying that no PdO phase can be detected.

This can be explained as follows. When the Pd(II) was introduced into the precursor solution, it would be coordinated with Thiourea (TU) molecules to keep stable in the solution. The coordination complex can be written as Pd(TU)₄Cl₂ (10.1016/j.molstruc.2005.01.008; 10.1007/s11243-008-9177-5), in which the Pd(II) is coordinated with four S atoms of TU molecules. For other metal elements such as Cu, Zn, they could have the similar state. When the precursor film was prepared and annealed at 280 °C in air, the main reaction is the solvent volatilization and partial decomposition of these TU coordination complex into metal sulfides. Due to the protection of S coordination, the Pd may not be very easy to sufficient contact with oxygen and transform into Pd oxides. This is reasonable since other metal oxides has

neither be observed although they have much higher concentrations.

Based on the above discussion, we think the formation of Pd oxides in the precursor film could be ignored here and the PdSe_x was mainly transformed from the selenization of Pd-TU coordination complex or its decomposition products.

Comment 8: During the high-temperature selenization process, Pd elements exhibit a higher diffusion rate in the absorber layer. Please explain why do Pd elements aggregate on the surface of the absorber layer? Additionally, Pd elements are mainly distributed at grain boundaries. In the lower layer of the absorber layer with smaller grains, there are more grain boundaries, theoretically suggesting a higher content of Pd elements.

Reply: Thanks very much for this comment. Regarding the appearance of Pd on the film surface, in our opinion, Pd has a high activity to react into PdSe_x compounds. In this case, the film surface could provide adequate space for the formation of PdSe_x phase. This is a common phenomenon in the selenization crystallization of CZTSSe materials. The metal elements with higher reaction activity with the Se vapor will spontaneously diffuse to the surface and crystallize there. The CZTSSe crystal surface can form very quickly in the selenization process. Since Pd cannot dope into the lattice, it must be segregated outside the CZTSSe grains, on the surface, in the grain boundary or in the bottom amorphous regions.

Regarding the location of Pd in the bottom fine grain region, we totally agree with you that these regions have more GBs and thus can contain more Pd. We have further measured the SIMS of the film, as shown above. It can be seen that in the bottom region

of the CZTSSe layer, the SIMS intensity of the PdSe₂⁺ exhibited an increase trend, which could be an indication of the accumulation of Pd in this region.

Comment 9: Does the reduction carrier recombination losses at heterojunction interfaces relate to an increase in the bandgap at the absorber layer surface or the easier promotion of carrier transport due to band bending at the absorber layer surface? Would a single gradient distribution of Pd elements alter the original band structure of the absorber layer?

Reply: Thanks very much for this comment. According to our DFT calculations, XRD, Raman and valence-band XPS results, we confirmed that the Pd doping into the CZTSSe lattice was negligible and no change in the valence band DOS can be observed. As such, we do not prefer to consider the increased bandgap at the absorber layer surface.

The change in the band bending can indeed occur. As measured by DLCP and CV, the carrier density of the Pd-ACZTSSe film has been increased. This will enhance the heterojunction built-in potential and electric field in the cell, thus enhancing the carrier transport and transfer at the interface and reducing the recombination.

Since Pd cannot dope into the lattice, its influence on the band structure can also be ignored here. Our XPS valence band spectra in Supplementary Figure 12 also demonstrated that the DOS of the material has not been changed.

Comment 10: ACZTSSe-Pd has achieved a high certification efficiency. The absorber layer material has evolved from the quinary CZTSSe to the septenary ACZTSSe-Pd, making bulk defects already very complex. Please properly propose a future research direction related of cationic doped CZTSSe for better device performance.

Reply: Thanks very much for this comment. As you suggested, we have added a discussion in last section as shown below. We think cation incorporation is promising and we still need to further understand the underlying material physical mechanism of the CZTSSe crystallization, defect formation and the role of the cations.

“Our study here and previously reported works have widely demonstrated the

promising role of cation incorporation in enhancing the performance of Kesterite solar cells. However, the introduction of a variety of cations also complexes this material system and causes controversies about the material physical mechanisms through which these cations exert their effects, alloying, doping or others. This demands us to better understand the crystallization growth and defect formation processes in the CZTSSe material. In particular, we need to pay more attention to the effects of these incorporated cations on the initial and intermediate states of the CZTSSe material, rather than just the semiconducting and defective properties of the final state material. These efforts will help us more effectively determine the current issues of CZTSSe materials and the related origins and thus guide us more synergistically exploit the positive role of these cations.”

Reply to reviewer 3

Firstly, we would like to show our sincere appreciation to your kind and professional comments and the recognition to our work. According to these comments and suggestions, this manuscript has been carefully revised. In the following, we will reply to the raised questions, suggestion or comments one by one.

1. Introduction: secondary phases are a major problem in the synthesis process of kesterites. The introduction mentions this topic only marginally and as a relevant issue at grain boundaries (GBs). While the presented study focuses on the effect of GBs, it should be clearly presented that the GBs are not the only reason for the low efficiency values obtained so far. Thus, a brief introduction to the impact of secondary phases on the solar cell performance should be included, to put the main aspect of the GB effects into perspective.

Reply: Thanks very much for your valuable comment. We totally agree with you that secondary phases are a very important issue associated to the fabrication of high-quality CZTSSe. We have revised the second paragraph of the introduction to give a more comprehensive description of the factors that influence the performance of CZTSSe solar cells.

“For polycrystalline CZTSSe, the complex phase evolution processes, the coexistence of secondary phases, and the disorder of multinary elements are considered as main causes of defects and charge loss. In previous studies, a variety of efforts have been paid to relieve these issues and particularly to suppress intrinsic point defects within the bulk grain interiors (GI), which has made considerable contribution to the efficiency improvement of the cell. In addition to these widely concerned issues, Hao et al. recently highlighted that the grain boundary (GB) within CZTSSe absorbers actually played a more substantial role in influencing charge recombination velocity and charge loss.”

2. The abstract states 14.5% efficiency, while the last paragraph of the introduction states 14.3% efficiency. I suppose this is because one is the lab measurement and the

second is the certified value. Nevertheless, it would be preferable to be consistent in the reporting of efficiency values.

Reply: Thanks very much for your kind remind. As you suggested, we have revised the abstract and used the certified 14.3% efficiency in this section.

“As a result, high-performance Kesterite solar cells with a total-area efficiency of 14.5% (certified at 14.3%) have been achieved.”

3. XPS data (Suppl. Fig. 2) is shown to support the presence of PdSe₂ on the surface. The XPS spectra shown only indicate the Pd 3d(5/2) peak, and the peak on the surface appears to be slightly asymmetric. Can the authors make any conclusion on the binding state from these data? Is the Pd peak shifted corresponding to PdSe₂ bonds? Are reference measurements of a PdSe₂ compound available to substantiate the conclusion toward a PdSe₂ compound?

Reply: Thanks very much for your comment. We are very sorry we have not clearly shown this point in the manuscript and the supplementary figure.

Here, Pd XPS spectra of the final film etched at different depth was measured to confirm the existence of PdSe_x covering on the film surface. In this figure, from the position of the Pd 3d peak, we think that the Pd was in a mixed state, +2 and +4, and that +2 Pd had a higher XPS intensity. In our measurement, no reference measurement was used because we were able to determine the XPS peaks corresponding to Pd²⁺ (~337 eV) or Pd⁴⁺ (338 eV) according to the evolution of Pd XPS spectra of the film at different selenization stages (Figure 4a).

For clarity, as shown in the above figure, we have revised this supplementary figure and fitted these spectra by dual peaks corresponding to Pd²⁺ and Pd⁴⁺.

4. Lines 105-109: With the addition of low ($\leq 3\%$) Pd concentrations, Pd doping would likely not be observed in XRD. Thus, the conclusion that Pd is not present in grains and only segregates to GBs is not sufficiently substantiated with the shown absence of an XRD shift.

Reply: Thanks very much for this comment. We agree with you that the unchanged XRD peak of the film under low-ratio addition ($\leq 3\%$) cannot completely exclude the possibility of doping. Firstly, we need to note that we did not use higher Pd addition in experiment because more addition would make the precursor solution unstable. Thus, most of our studies and conclusions were based on the low-ratio addition. In principle, significantly increasing the addition amount will enhance the doping probability despite of the large doping formation energy.

Now, let us look back at our experimental data. First are the Raman spectra. As shown in the below figure, no shift of CZTSSe Raman peak can be observed when increasing the Pd ratio. If Pd has effectively doped into the lattice, due to difference in the electronic structure between Pd and other metal elements and difference in the metal-Se interactions, the lattice vibration properties as well as the Raman peak should be changed. As such, from the Raman spectra, we conclude that effective doping of Pd into the CZTSSe lattice did not occur in our experiment.

Second are the XPS valence-band spectra, as shown in the below figure. We have calculated the influence of Pd doping on the DOS of CZTSSe. It can be found that doping Pd will introduce high electronic state in the valence band edge. However, we had not observed this additional state in the XPS valence-band spectra. This result also

helps us to exclude the effective doping of Pd in CZTSSe lattice.

Overall, these experimental and theoretical data make us get the conclusion that Pd was mainly segregated to the GB regions. Of course, we cannot completely exclude the possibility of a very low doping since this can hardly be detected due to their little influence on the lattice parameter, vibration and electronic properties. Nonetheless, as you suggested, to avoid overly arbitrary statements, we have revised the conclusion of this part as “**Therefore, we speculate that the doping of Pd atoms into the Kesterite lattice is negligible.**”

5. Supplementary Fig. 4: why were the stated chemical potentials selected? What is their significance?

Reply: Thanks very much for this question. We are very sorry for not describing the details of chosen chemical potentials clearly. The stability of target semiconductor $\text{Cu}_2\text{ZnSnSe}_4$ (CZTSe) needs element chemical potentials satisfying some conditions: (i) the sum of chemical potentials of component elements equals the formation enthalpy of the compounds; (ii) the pure elemental phases of all component elements should not form; (iii) the formation of the secondary compounds is avoided. Following these constraints, the stable chemical potential region of CZTSe can be determined, as shown in the below figure. Since CZTSe is quaternary compound, there are three independent chemical potential variables, μ_{Cu} , μ_{Zn} , and μ_{Sn} , thus the stable region of CZTSe is limited in the 3-dimensional space. The element chemical potentials in such region stabilize the

formation of CZTSe against the co-existence of secondary phases, such as CuSe, CuSe₂, SnSe, SnSe₂, ZnSe, Cu₃Se₂ and Cu₂SnSe₃. The chemical potentials chosen in E-H in the supplementary figure are some typical endpoints of the 3-dimensional stable region.

To exhibit the stable region more clearly, we choose other 2-dimensional planes, in which the chemical potentials of Cu are fixed at -0.2007 eV, -0.2507 eV, -0.3307 eV and -0.6000 eV, respectively. The corresponding 2-dimensional stable regions are shown in gray area in the below figure. The chemical potentials chosen in A-D in the supplementary figure are representative points in these stable regions. Since in this calculation Pd is considered as the dopant in the CZTSe sample, the chemical potential of Pd should satisfy the extra condition that all the Pd-related secondary phases, such as Cu₃Pd, PdSe₂, SnPd, SnPd₂ and SnPd₃, cannot form. Based on the chemical potentials of Cu, Zn, Sn, Se obtained above, the chemical potentials of Pd can be then determined.

Fig. The calculated chemical-potential stability diagrams (gray area) of $\text{Cu}_2\text{ZnSnSe}_4$ in 2D planes with different μ_{Cu} .

The chemical potentials chosen in supplementary fig. 6 represent different synthesized conditions of CZTSe. The more negative value of element chemical potential means the element is poorer. As we see, the chosen value roughly spreads the variation range of element chemical potential in 3-dimensional space, covering most probable growth conditions in experiment. No matter how the chemical potentials change, the formation energies of Pd-related defects are always higher than those of intrinsic defects, indicating that the Pd impurities are more difficult to form in CZTSe lattice, consistent with our experimental observations.

For clarity, we have added a brief of this information in the DFT calculation section as “For the chemical potentials used in the defect formation calculation, they were selected from the calculated phase diagram of CZTSe, including typical endpoints of the Cu-Zn-Sn 3-dimensional stable phase region and some representative points at different Cu chemical potentials.”

6. Line 112: It is claimed that the formation energy for Pd substitution in specific sites in the kesterite lattice is larger than 1 eV. What is the formation energy for a Pd substitution in the GB? How does it compare?

Reply: Thanks very much for your question. Honestly, we did not perform this DFT calculation. During our initial study in this work, we had discussed this topic with our theoretical team (Prof. Shiyu Chen). We found that the atomic structure of surface and GB of CZTSSe is too complex to be theoretically constructed for the DFT calculation since no sufficient experimental information regarding these regions has been obtained until now. This means that our theoretically constructed atomic structure is likely to be inconsistent with the actual situation, thus leading to unreliable results.

From the perspective of CZTSSe crystallization and growth, we think it is possible for Pd to dope at the surface or GB regions during the high temperature selenization process since these regions usually have large structural distortion in this stage. However, when

the crystal ripening gradually occurs and the reaction temperature gradually decreases, structural reconstruction of these regions will occur, which could push out the already-doped Pd.

Overall, study on the atomic structure of surface and GB of CZTSSe materials from both experiment and theoretical calculations is worthy of further exploration in the future.

7. I would like to encourage the authors to rethink the use of colors in Fig. 1. Some of the panels consistently use blue and red for ACZTSSe-Pd and control samples (as done also in other Figures), which is very helpful to the reader. Other panels (such as (d), (d) and (f) are not consistent with this choice. Please consider using red and blue tones in (e) and (f), similar to (h) and (i). Hopefully also (d) can be changed to be consistent in the use of colors.

Reply: Thanks very much for your kind remind. We totally agree you that consistent color form will help the reader to follow our presentation. As suggested, we have thoroughly revised these figures.

8. Do the data in Supplementary Fig. 7 correspond to the final film after 20 minutes selenization time? It would be helpful to state this specifically in the caption.

Reply: Thanks very much for your comment. We are very sorry we have not clearly given this information in the figure caption. The XPS spectra in this figure is measured on the final films. We have revised the figure caption as “**Supplementary Figure 9. XPS spectra of (A, E) Cu 2p_{3/2}, (B, F) Ag 3d_{5/2}, (C, G) Zn 2p_{3/2} and (D, H) Se 3d of final-state control and ACZTSSe-Pd films (A-D: Control, E-H: ACZTSSe-Pd).**”

9. Is the data shown in Supplementary Fig. 10 from DFT or experimental data? This should be clearly stated in the caption.

Reply: Thanks very much for this question. We are very sorry we have not clearly given this information in the figure caption. It is experimental data. As suggested, we have revised the figure caption as “**Supplementary Figure 12. The measured XPS valence-**

band spectra of (A) control and (B) ACZTSSe-Pd films selenized for 20 min. The dashed black lines mark the baseline and the tangents of the curve. The intersections of the tangents with the baseline give the valence band maximum position vs Fermi energy.”

10. Regarding the KPFM data shown in Fig. 1 (H) and (I), it is essential to know how the measurements were done and how the CPD is defined. Is it work function of tip minus that of sample, or the opposite? Only with this information, the conclusion toward stronger p-type of the Pd sample can be made. This information should be clearly stated in the experimental section. Furthermore, comparing CPD values from sample to sample requires the use of the same tip without any tip modification (e.g. due to accidental tip crashes). Was this assured in the presented experiments?

Reply: Thanks very much for your valuable comment. We totally agree with you that we need to clearly give the information regarding how the CPD was defined in our measurement.

In our measurement, the CPD is the potential difference between the sample and the probe tip, that is, $CPD = \phi_{\text{sample}} - \phi_{\text{tip}} = (W_{\text{tip}} - W_{\text{sample}})/e$, where ϕ is the electric potential, W is the work function and e is the elementary charge. Thus, lower CPD refers to higher work function and more obvious p-type characteristics. As suggested, we have added this information in the experimental section for clarity as “For the KPFM, the CPD was measured as the potential difference between the sample and the probe tip, that is $CPD = \phi_{\text{sample}} - \phi_{\text{tip}} = (W_{\text{tip}} - W_{\text{sample}})/e$, where ϕ is the electric potential, W is the work function and e is the elementary charge.”

We totally agree with you that in the CPD comparison study between different samples, we should keep the tip in the same condition. In our experiment, we used the same tip for the sequential measurements of different samples. The measurements were completed in a relatively short time duration. We have performed carefully to avoid touching or defacing the probe tip in this process. Additionally, if the probe tip was defaced, no clear CPD images could be obtained. These operation and phenomena helped us assure the reliability of this measurement.

11. Supplementary Fig. 11: The authors do not comment on the clear contrast at the GBs for control and ACZTSSe-Pd samples. For the latter, this contrast is very clear, while it is weaker for the control sample. Interestingly, the contrast at the GBs seems to invert for both samples after the 20 min selenization. In the main manuscript in Fig. 2, only the 20 min images are shown and discussed. Since the contrast inverts for these images, a proper analysis and discussion needs to be included to why this might happen and why the GBs at earlier stages in the selenization process show the opposite band bending.

Furthermore, regarding Fig. 2, the figure caption or the images (I) through (L) should indicate which sample is shown. Additionally, the caption indicates that (K) and (L) correspond to KPFM images, while they are obviously schematic drawings of the potential profile at the GBs. The caption should be corrected accordingly.

Reply: Thanks very much for this comment and your careful reviewing. We are very sorry we have not clearly show these CPD images because we initially wanted to use the same color bar in the supplementary figure. Here, we have changed the figure color bar and added topography and potential line scans to make the contrast more clearly be seen.

Supplementary Figure 13. AFM and KPFM images and topography and CPD line scans of control films selenized at (a,e,i) 4 min, (b,f,j) 6 min, (c,g,k) 10 min and (d,h,l) 20 min. AFM and KPFM images and topography and CPD line scans of ACZTSSe-Pd films selenized at (m,q,u) 4 min, (n,r,v) 6 min, (o,s,w) 10 min and (p,t,x) 20 min. For the control sample, in the first 10 mins selenization, lower CPD was obtained in the GB regions, similar to that of the ACZTSSe-Pd sample. However, due to the Se loss in the later-stage high temperature process, a significant amount of V_{Se} would appear in the GB regions of the control sample, introducing donor charge, thus causing the contrast

inverts of the CPD in the GB regions. Comparatively, the ACZTSSe-Pd sample did not show this phenomenon because the element loss has been effectively suppressed.

Regarding the evolution of CPD during the selenization process, we have also added a discussion in this supplementary figure caption as “For the control sample, in the first 10 mins selenization, lower CPD was obtained in the GB regions, similar to that of the ACZTSSe-Pd sample. However, due to the Se loss in the later-stage high temperature process, a significant amount of V_{Se} would appear in the GB regions of the control sample, introducing donor charge, thus causing the contrast inverts of the CPD in the GB regions. Comparatively, the ACZTSSe-Pd sample did not show this phenomenon because the element loss has been effectively suppressed.”

Regarding why in the early stage, the GB region had lower CPD, honestly, we still do not know the exact reason. Lower CPD means more obvious p-type doping. Maybe the GB region has large degree of structural distortion, leading to more significant Cu_{Zn} acceptor defects. Another possible cause could be Na induced p-type doping in this region since it was found that Na doping mainly existed in the GB region.

As suggested, we have also revised the caption in figure 2 as “(I-J) KPFM mapping images of the two absorber films (I: Control, J: ACZTSSe-Pd) and (K-L) schematic diagram of the energy band bending near the GB regions (K: Control, L: ACZTSSe-Pd). “e⁻” represents electron and “h⁺” represents hole.”

12. Supplementary Fig. 14: It is not really clear from the images shown that there is a difference between the control sample and the ACZTSSe-Pd sample. What specific features do the white arrows point to? It is not clear how they are different between control and Pd sample.

Reply: Thanks very much for your comment. We are very sorry we have not clearly described this figure. These SEM images and EDX mapping results were mainly used to give a qualitative observation of Se or Sn element deficiency in the GB regions. In this figure, from the qualitative perspective, we can see the deficiency of Se and Sn in

some GB regions of the control sample (as in B-C), while for the ACZTSSe-Pd sample, no such deficiency phenomena were observed (H-I). To more clearly describe this result, we have revised the figure caption as “**Supplementary Figure 16. (A) Top-view SEM image and (B-F) EDX mapping of the control sample (B: Se, C: Sn, D: Cu, E: Zn, F: Ag). (G) Top-view SEM image and (H-L) EDX mapping of the ACZTSSe-Pd film (H: Se, I: Sn, J: Cu, K: Zn, L: Ag). GBs are marked with yellow dotted lines. It can be qualitatively seen in (B-C) in some GB regions Se and Sn exhibited obvious deficiency, as depicted by the white arrows, while for the ACZTSSe-Pd sample in (H-I), no obvious Sn or Se deficiency was observed in the GB region.**”

Supplementary Figure 16.

13. For the solar cell fabrication, was the PdSe₂ which is observed at the absorber surface removed in any way? PdSe₂ is reported to be a low bandgap semiconductor (for more than single layer PdSe₂), which should be expected to be detrimental for PV performance, if present at the pn-junction.

Reply: Thanks very much for this valuable comment. In our experiment, we did not carry out any additional procedure to intentionally remove the residual PdSe_x compounds. Nonetheless, we found that the content of Pd on the film surface could be

significantly reduced by immersing the film in the ammonia solution that used for the CdS CBD process. As shown in the below figure, the Pd XPS intensity was reduced by more than a half after a 15 min solution etching. As such, it can be speculated that during the CdS CBD process, the Pd at the CZTSSe/CdS interface can be obviously removed, which could reduce their possible negative influence on the cell performance. Also, as you suggested, in the future we may need to explore more other method to more effectively remove these detrimental phases.

14. More details about the CdS recipe should be provided.

Reply: Thanks very much for this suggestion. As suggested, we have added more detailed information regarding the CdS recipe and deposition process in the experimental section.

“Specifically, ACZTSSe films were immersed into the aqueous solution (200 ml water) pre-dissolved CdSO₄ (~12 mM) and ammonia (~10 ml) in a beaker. Thiourea (~48 mM) was then added into the solution. After the thiourea was completely dissolved, the beaker was immersed into the 70 °C water bath to start the CdS deposition. The CBD was performed for about 10-11 min to get the desired thickness. These films were then cleaned by water and dried by N₂ for the following window layer deposition.”

15. Some specific corrections:

o Line 96: I suppose that it should state ACZTSSe-Pd, as the film shows the presence of Pd in the EELS data. Same in lines

o Line 107/108: The use of “neither” requires a “nor”. Is this sentence incomplete?

Reply: Thanks very much for your kind remind. We have revised these language errors in this manuscript.

REVIEWERS' COMMENTS

Reviewer #1 (Remarks to the Author):

The revised version shall be accepted.

Reviewer #2 (Remarks to the Author):

The manuscript has been significantly improved and could be published as it is.

Reviewer #3 (Remarks to the Author):

The authors presented a revised manuscript, in response the reviewer comments of three reviewers. First, I would like to acknowledge the clear and efficient presentation of the responses and changes to the manuscript, which facilitated the reviewing of the revised manuscript.

The authors addressed all my comments and suggestions, and the manuscript can be recommended for publication. I would suggest just a few further changes and one doubt/comment to be implemented prior to acceptance.

In lines 239-245, the authors about “interface defect-induced charge loss”, it is not clearly stated which interface is referred to here. Is it the grain boundary interface, or the interface of ACZTSSe with the buffer layer? Furthermore, the interface defect densities are given, extracted from DLCP and C-V experiments. The values are given with a unit cm^{-3} , a volume density. However, typically, interface defect densities are given per area, namely with a unit of cm^{-2} . Is this a typo? Was there any estimation of an “interface thickness” considered to reach densities per volume? In case that the interface densities are actually stated as per area, the values are quite large, compared to values that have been reported for related CIGSe solar cells (see for example Sadewasser et al., *Thin Solid Films* 431-432, 257 (2003); Fuertes Marrón et al., *Phys. Rev. B* 71, 033306 (2005)). Additionally, such high interface defect densities would likely lead to Fermi-level pinning. The limit for Fermi-level pinning was discussed by Rau et al., *J. Appl. Phys.* 86, 497 (1999)).

Furthermore, I would like to suggest some minor language corrections to improve readability:

Line 29, correct to “contributing to eliminating the already-existing ...”

Line 79, correct to "advantage" instead of "advantaging"

Line 400, add "performed" to read "...of the device was performed before the measurement".

Reply to reviewer 3

Firstly, we would like to show our sincere appreciation to your positive recognition to our revisions. For the new raised comment, we will reply in the following.

1. In lines 239-245, the authors about “interface defect-induced charge loss”, it is not clearly stated which interface is referred to here. Is it the grain boundary interface, or the interface of ACZTSSe with the buffer layer? Furthermore, the interface defect densities are given, extracted from DLCP and C-V experiments. The values are given with a unit cm^{-3} , a volume density. However, typically, interface defect densities are given per area, namely with a unit of cm^{-2} . Is this a typo? Was there any estimation of an “interface thickness” considered to reach densities per volume? In case that the interface densities are actually stated as per area, the values are quite large, compared to values that have been reported for related CIGSe solar cells (see for example Sadewasser et al., *Thin Solid Films* 431-432, 257 (2003); Fuertes Marrón et al., *Phys. Rev. B* 71, 033306 (2005)). Additionally, such high interface defect densities would likely lead to Fermi-level pinning. The limit for Fermi-level pinning was discussed by Rau et al., *J. Appl. Phys.* 86, 497 (1999)).

Reply: Thanks very much for your valuable comment. We are very sorry we have not clearly described this information in the manuscript.

The “interface” here is referred to the ACZTSSe/CdS interface according to the spatial profiling by DLCP and CV. The interface defect density (N_{IT}) extracted from DLCP and C-V experiments primarily represents the volume defect density of the region near the ACZTSSe/CdS interface (*J. Appl. Phys.* 95, 1000 (2004)), which uses the cm^{-3} as the unit. This method has been widely used in this field (*Adv. Energy Mater.* 2014, 4, 1301465; *Nat. Energy* 2022, 7, 966-977). We agree with you that the simple naming “interface defect” may introduce misunderstanding. To address this issue, in the revised manuscript, we used “volume defect density near the ACZTSSe/CdS interface region” to replace the original description of “interface defect density”.

If wanting to use cm^{-2} as the unit, the thickness (x) of the film should be involved in the calculation, that is multiplying N_{IT} and x . Taking the ACZTSSe-Pd sample as an

example, the N_{IT} and x are about $4.5 \times 10^{15} \text{ cm}^{-3}$ and 2 μm , respectively, and thus their product is about $9 \times 10^{11} \text{ cm}^{-2}$, which is comparable to the value ($\sim 10^{11}$ and 10^{12} cm^{-2}) reported in the literatures (Thin Solid Films 431, 257 (2003); Phys. Rev. B 71, 033306 (2005)).

In our previous works, we also developed a method to derive the interface defect density per area (Adv. Energy Mater. 2019, 9, 1901352). Here we also used this method to estimate the interface defect density of the cell and to make a comparison between different methods. As shown in the figure below, through fitting the frequency-dependent admittance curve, the density of interface defects that participated in the charge capturing in the ACZTSSe-Pd sample is estimated to be $\sim 3.5 \times 10^{12} \text{ cm}^{-2}$, which had a comparable order of magnitude to the value extracted from DLCP and C-V experiments as well as reported in above literatures.

2. Some specific corrections:

Line 29, correct to “contributing to eliminating the already-existing ...”

Line 79, correct to "advantage" instead of “advantaging”

Line 400, add “performed” to read “...of the device was performed before the measurement”.

Reply: Thanks very much for your kind remind. We have revised these language errors in this manuscript.